# Sex and organ specific proteomic responses to vitamin C deficiency in the brain, heart, liver, and spleen of *Gulo*-/- mice

**Lucie Aumailley, Michel Lebel**[ID]*

Centre de Recherche du CHU de Québec, Faculty of Medicine, Université Laval, Québec City Québec, Canada

* michel.lebel@crchudequebec.ulaval.ca

## Abstract

Recent advances in mass spectrometry have indicated that the water-soluble antioxidant vitamin C differentially modulates the abundance of various proteins in the hepatic tissue of female and male mice. In this study, we performed LC-MS/MS to identify and quantify proteins that correlate with serum vitamin C concentrations in the whole brain, heart, liver, and spleen tissues in mice deficient for the enzyme L-Gulonolactone oxidase required for vitamin C synthesis in mammals. This work shows for the first time that various biological processes affected by a vitamin C deficiency are not only sex specific dependent but also tissue specific dependent even though many proteins have been identified and quantified in more than three organs. For example, the abundance of several complex III subunits of the mitochondrial electron transport chain correlated positively with the levels of serum vitamin C only in the liver and not in the other tissues examined in this study even though such proteins were identified in all the organs analyzed. Western blot analyses on the Uqcrc1 and Uqcrfs1 complex III subunits validated the mass spectrometry results. Interestingly, the ferritin subunits represented the few quantified protein complexes that correlated positively with serum vitamin C in all the organs examined. Concomitantly, serum ferritin light chain 1 was inversely correlated with vitamin C levels in the serum. Thus, our study provides an initial comprehensive atlas of proteins significantly correlating with vitamin C in four organs in mice that will be a useful resource to the scientific community.

## Introduction

Recent advances in high-throughput technologies in proteomics have substantially increased our knowledge on tissue specific molecular regulators and effectors of physiological activities in different organs [1–4]. Deep proteomic analyses provide useful organ specific catalogues or atlas of protein expression and networks that can be thoroughly examined in various pathophysiological conditions [5].

Several epidemiological studies have indicated that large subpopulations (between 5% and 30%, depending on socioeconomic status, smoking habit, and age) can be diagnosed with

**Data Availability Statement:** All mass spectrometry data (raw files and MaxQuant search result files) are publicly available on ProteomeXchange repository (www.proteomexchange.org) with the identifier

MSV000092125 for the brain samples, MSV000092122 for the heart samples, MSV000092040 for the liver samples, MSV000092120 for the spleen samples, and PXD027019 for the serum samples.

**Funding:** This work was supported by the Canadian Institutes of Health Research, Canada (PJT-173398) to Michel Lebel. The funders had no role in study design, data collection and analysis, decision to publish, or preparation of the manuscript.

**Competing interests:** The authors have declared that no competing interests exist.

hypovitaminosis C [6–8]. Even though a hypovitaminosis C condition (or low blood levels of vitamin C) may not lead to scurvy, it places an individual at higher risk for metabolic abnormalities, cardiovascular diseases, and cancer [9–11]. Clinically relevant animal models of vitamin C deficiency like the *Gulo*[-/-] mice are essential for improving our understanding of the role of vitamin C in normal physiology [12–16]. By simply removing vitamin C from drinking water, all organs will exhibit a significant decrease of this vitamin after the first week of withdrawal [17]. In doing so, several studies on vitamin C deficient *Gulo*[-/-] mice have indicated that low levels of vitamin C increase aortic wall damage and sensorimotor deficits, induce impaired neutrophil apoptosis and clearance, deteriorates bone microarchitecture, and decrease the life span of these mice [12–16].

In a previous study with *Gulo*[-/-] mice, we performed label-free Liquid Chromatography-Tandem Mass Spectrometry (LC-MS/MS) to quantify proteins from microsomal enriched liver extracts to identify the proteins that correlated with the hepatic concentrations of vitamin C [18]. Although a sexual dimorphism could be detected with the proteins that correlated positively or inversely with hepatic vitamin C concentrations, both males and females exhibited an increase in several factors associated with the classical pathway of complement activation (immune innate system) and a decrease of several proteins of the mitochondrial complex III of the electron transport chain upon vitamin C withdrawal from drinking water [18]. Concomitantly, vitamin C deficient *Gulo*[-/-] mice showed a decrease in mitochondrial complex III activity, lower ATP production, and increased oxidative stress. However, it is unknown whether the vitamin C deficient induction of mitochondrial dysfunction is liver specific or is also observed in other organs. Similarly, it is unclear whether common biological processes are affected by a vitamin C deficiency in various organs. In this study, we performed LC-MS/MS to identify and quantify proteins that correlate with serum vitamin C concentrations in the whole brain, heart, liver, and spleen tissues. A high percentage of brain and heart tissues are composed of post-mitotic cells. The liver is composed of mainly hepatocytes with regenerative potential. The spleen possesses cells with highly proliferative capacity. As such this study allowed us to compare the proteomic profiles of tissues with different energetic demands. This work reports for the first time that the biological processes affected by a vitamin C deficiency are not only sex specific dependent but also tissue specific dependent even though many proteins have been identified and quantified in more than three organs. Finally, the abundance of several complex III subunits of the mitochondrial electron transport chain is altered only in the liver and not in the other tissues examined in this study. In contrast, the abundance of the ferritin complex correlates with serum vitamin C levels in all the organs examined. Concomitantly, serum ferritin is inversely correlated with serum vitamin C concentrations.

## Materials and methods

### Animals and maintenance

*Gulo*[-/-] mice were obtained from the Mutant Mouse Regional Resource Centers (University of California Davis, CA) and were housed at the Centre Hospitalier de l'Université Laval animal facility and maintained with 0.4% (w/v) of L-ascorbate (vitamin C; Sigma-Aldrich, Oakville, ON) in drinking water. These mice were backcrossed onto the C57BL/6NHsd background (Harlan Laboratories, Frederick, MD) for 12 generations. Finally, heterozygous mice were crossed to obtain *Gulo*[-/-] and wild type (*Gulo*[+/+]) mice. This study was carried out in strict accordance with the recommendations in the Guide for the Care and Use of Laboratory Animals of the Canadian Council on Animal Care in science and the protocol was approved by the Committee on the Ethics and Protection of Animal of Laval University (Permit Number: CHU-22-1039). Mice were housed in cages (containing a top filter) at $22 \pm 2°C$ with 40%–50%

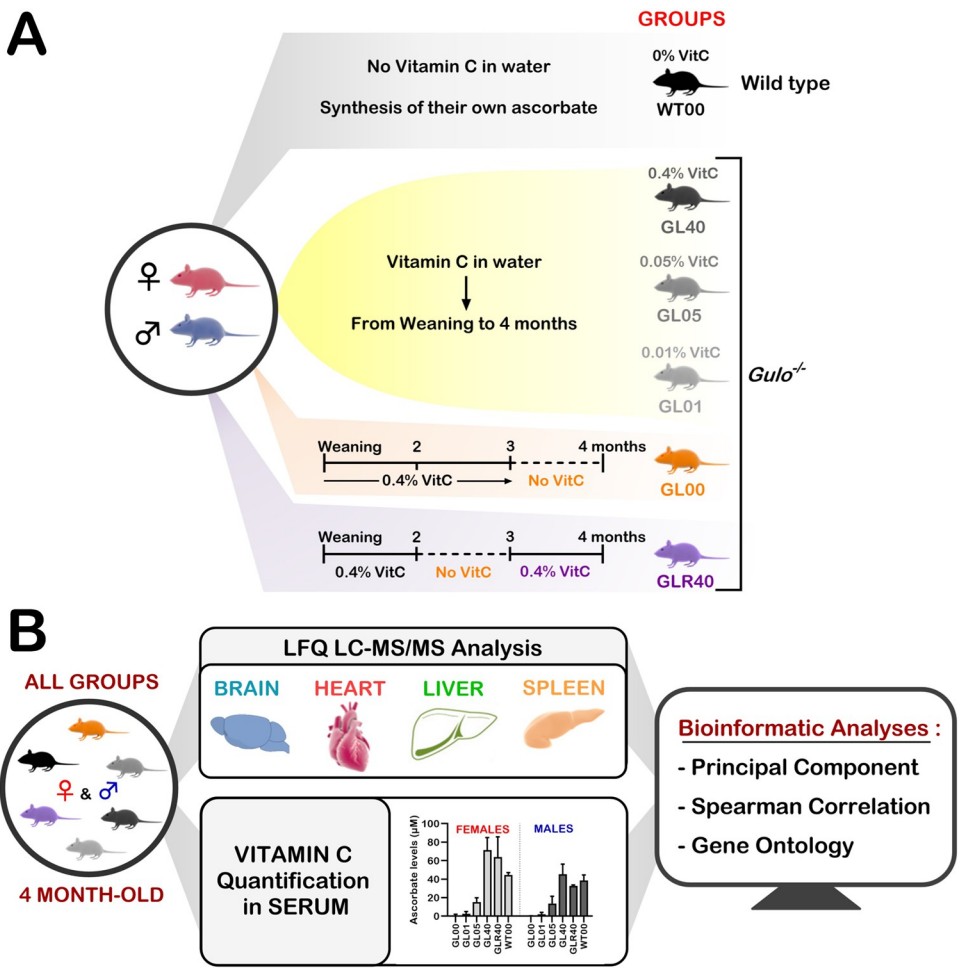

**Fig 1. Experimental study design.** (A) Schematic of the different vitamin C treatments of mice. Females and males were separated into six experimental groups. Animals were labeled according to their genotype (GL for *Gulo*−/− mice and WT for wild-type mice) and the vitamin C treatments (% is weight of vitamin C per 100 mL of drinking water). The timelines for each treatment are indicated. (B) Overview of the research design. Brain, heart, liver, and spleen were harvested at the age of 4 months for all mice. Label free quantification LC-MS/MS analyses were performed on the tissue samples as well as measurements of serum vitamin C levels. Principal component analysis, Spearman rank correlation evaluation, and gene ontology study of the omics data led us to the identification of proteins and biological processes that were modulated by serum vitamin C levels in the different organs.

humidity and a 12-h light–dark cycle (light cycle: 07:00–19:00 hours). All mice were fed ad libitum with Teklad Global 18% protein rodent diet, 6% fat, 110 IU/kg of vitamin E, and 15 IU/g of vitamin A (Envigo cat. # 2918, Madison, WI).

Mice were separated into six cohorts containing three males and three females each (see Fig 1A for summary). One cohort of *Gulo*-/- mice was maintained on standard diet and supplemented with 0.4% ascorbate (w/v) in drinking water from weaning until the age of four months (referred as GL40). A second cohort of *Gulo*-/- mice was treated with 0.05% ascorbate in drinking water from weaning until the age of four months (GL05). A third cohort of *Gulo*-/- mice was supplemented with 0.01% ascorbate from weaning until the age of four months (GL01). A fourth cohort of *Gulo*-/- mice was treated with 0.4% ascorbate until the age of three months. Ascorbate was then removed from drinking water for four weeks (GL00). Mice were not kept beyond four weeks without ascorbate in drinking water as they were losing more than

15% of their body weight and became moribund [12]. A fifth cohort of *Gulo*$^{-/-}$ mice were treated with 0.4% ascorbate from weaning until the age of two months. Ascorbate was removed from drinking water for one month. Then ascorbate was added back (0.4%) to drinking water until the mice reached the age of four months (GLR40). This cohort was considered the vitamin C rescue cohort. Finally, wild type control (*Gulo*$^{+/+}$) mice were maintained in the same room with no ascorbate supplementation in drinking water and were used as our normal reference cohort (WT00) (Fig 1A).

## Organ collection from the different mouse cohorts

Mice were fasted overnight before organ collection. Brain, heart, liver, and spleen were harvested at 10:00 am the next day after final exsanguination under general anesthesia (with 3% isoflurane) at the age of four months. Organ samples were frozen at -80˚C until further analyses.

## Vitamin C (ascorbate) quantification in serum samples

Blood was allowed to clot for one hour on ice and spun on a bench top centrifuge at 16,000 *g* for 15 min. Collected serum was frozen in aliquots at -80˚C until execution of analyses. Vitamin C in serum was measured with the ferric reducing ascorbate assay kit from BioVision Research Products (Mountain View, CA, USA) as described previously [12].

## Whole tissue protein extraction

Tissue protein extraction was carried out in lysis buffer containing 50 mM Tris-HCl (pH 7.5), 150 mM NaCl, 1% NP-40, 0.2% SDS, 1% sodium deoxycholate, 1mM phenylmethylsulfonyl-fluoride, complete protease inhibitor cocktail and phosphatase inhibitor cocktail PhoSTOP$^{TM}$ (Roche Applied Science, Indianapolis, IN, USA). After tissue homogenization and sonication, samples were centrifuged at 16,000 *g* for 15 min thrice. Protein concentration was determined by the Bradford protein assay (Bio-Rad, Mississauga, ON, Canada). Samples were frozen at -80˚C until mass spectrometry analysis.

## Preparation of sample for label-free Liquid Chromatography-Tandem Mass Spectrometry

Preparation of protein samples for mass spectrometry analysis were performed as described in previous studies by Aumailley *et al.* [18, 19]. Briefly, each protein sample was precipitated with 5 volumes of acetone overnight. The protein pellet was recovered by centrifugation at 16,000 *g* for 15 min and resuspended in 200 μL of 1% DOC / 50 mM ammonium bicarbonate buffer. A Bradford protein assay was performed to estimate protein concentration. Ten μg of proteins from lysate were digested with trypsin. Briefly, proteins were first reduced with 0.2 mM dithiothreitol for 30 min at 37˚C and alkylated with 0.8 mM iodoacetamide for 30 min at 37˚C. Samples were then incubated with trypsin (trypsin:protein; 1:50) at 37˚C overnight. The reaction was stopped by addition of 1% trifluoroacetic acid (TFA), 0.5% acetic acid, and 0.5% acetonitrile then centrifuged for 5 min at 16,000 *g*. The peptides obtained were then desalted using C18 stagetip. Protein concentrations were measured with a peptide nanodrop assay.

## Label-free Liquid Chromatography-Tandem mass spectrometry analysis of the samples

One μg of each organ sample was analyzed by nanoLC/MSMS using a Dionex UltiMate 3000 nanoRSLC chromatography system (Thermo Fisher Scientific, San Jose, CA) connected to an

Orbitrap Fusion mass spectrometer (Thermo Fisher Scientific) equipped with a nanoelectrospray ion source. Peptides were trapped at 20 μL/min in loading solvent (2% acetonitrile, 0.05% TFA) on a 5 mm x 300 μm C18 pepmap cartridge pre-column (Thermo Fisher Scientific) for 5 min. Then, the pre-column was switched online with Pepmap Acclaim column (Thermo Fisher Scientific) 50 cm x 75 μm internal diameter separation column and the peptides were eluted with a linear gradient from 5–40% solvent B (A: 0,1% formic acid, B: 80% acetonitrile, 0.1% formic acid) in 90 min, at 300 nL/min for a total run time of 120 min. Mass spectra were acquired using a data dependent acquisition mode using Thermo XCalibur software version 4.1.50. Full scan mass spectra (350 to 1800 m/z) were acquired in the orbitrap using an AGC target of 4e5, a maximum injection time of 50 ms, and a resolution of 120,000. Internal calibration using lock mass on the m/z 445.12003 siloxane ion was used. Each MS scan was followed by acquisition of fragmentation MS/MS spectra of the most intense ions for a total cycle time of 3 s (top speed mode). The selected ions were isolated using the quadrupole analyzer in a window of 1.6 m/z and fragmented by Higher energy Collision-induced Dissociation (HCD) with 35% of collision energy. The resulting fragments were detected by the linear ion trap in rapid scan rate with an AGC target of 1e4 and a maximum injection time of 50 ms. Dynamic exclusion of previously fragmented peptides was set for a period of 30 s and a tolerance of 10 ppm.

## Database searching and Label Free Quantification (LFQ)

Searches in databases and quantification of spectra were performed as previously described [18]. Briefly, spectra were searched against the Uniprot Ref *Mus musculus* database (July 2020 release/ 63807 entries) using the Andromeda module of MaxQuant software v. 1.6.10.43 [20]. Trypsin/P enzyme parameter was selected with two possible missed cleavages. Carbamido-methylation of cysteins was set as fixed modification while methionine oxidation, protein N-terminal acetylation and hydroxyproline were set as variable modifications for the global search. Mass search tolerance was 5 ppm and 0.5 Da for MS and MS/MS, respectively. For protein validation, a maximum False Discovery Rate of 1% at peptide and protein level was used based on a target/decoy search. MaxQuant was also used for Label Free Quantification. The 'match between runs' option was used with 20 min value as alignment time window and 0.7 min as match time window. Only unique and razor peptides were used for quantification. Normalisation (LFQ intensities) was performed by MaxQuant.

## LFQ data post-processing and statistical analysis

RStudio 1.2.5019 was used for data post-processing. In the case of protein intensity values that were missing, values were replaced by a noise value corresponding to 1% percentile of the normalised value for each condition. A protein was considered as quantifiable only if at least two peptides were identified for this protein. Principal Component Analysis (PCA) was performed with ClustVis [21]. Spearman correlation was calculated using Perseus (MaxQuant, v2.0.7.0, Martinsried, Germany) [22].

## Immunoblotting analysis

Western blotting analyses on protein samples from the various organs and the serum were performed as previously described [18]. Selected proteins on the PVDF membranes were detected using the following antibodies: rabbit polyclonal antibodies against ferritin light chain 1 (anti-Ftl1 #10727-1-AP) and against S100a9 (anti-S100A9 #26992-1-AP) from Proteintech (Rosemont, IL, USA), mouse monoclonal antibodies against ubiquinol-cytochrome C reductase core protein I (anti-Uqcrc1 [16D10AD9AH5] ab110252) and against ubiquinol-cytochrome c

reductase, Rieske iron-sulfur polypeptide 1 (anti-Uqcrfs1 [5A5] ab14746) from Abcam (Cambridge, MA, USA). Coomassie blue staining was used as in-gel loading control. Signals were quantified using ImageJ software (1.48v; http://imagej.nih.gov/ij). The original uncropped and unadjusted blot/gel images can be found in the supporting information file entitled S1 Raw image.

## Results

### Experimental design

To identify the biological processes that are altered in whole tissues such as the brain cortex, the heart, the liver, and the spleen of mice exhibiting different degrees of vitamin C deficiency at the proteome level, specific cohorts of female and male mice were subjected to different vitamin C concentrations in drinking water. Fig 1A presents the different mouse groups treated with the indicated concentrations of vitamin C in drinking water. Succinctly, six different cohorts containing at least three females and three males were used in this study. *Gulo*$^{-/-}$ mice were treated from weaning until the age of four months with 0.4% (GL40), 0.05% (GL05) or 0.01% (GL01) vitamin C (w/v). One cohort of *Gulo*$^{-/-}$ mice underwent a four-week vitamin C depletion from the age of three months until the age of four months (GL00). Another cohort of *Gulo*$^{-/-}$ mice were supplemented with 0.4% vitamin C (w/v) from weaning to the age of two months after which they experienced a four-week vitamin C depletion. At the age of three months, 0.4% vitamin C (w/v) was added back (for rescue) to drinking water until the age of four months (GLR40). The wild type (WT00) control mice, which synthesize their own vitamin C, were used as a normal reference cohort without vitamin C supplementation in drinking water. Using a ferric reducing ascorbate assay kit, we had measured vitamin C levels in the serum of each mouse in the different treated groups in a previous study [19]. The raw data of such measurements are shown in S1 Table. The lowest levels of serum vitamin C were recorded in the GL00 mice (males and females) and the highest levels were measured for the GL40 and GLR40 mice. The levels of serum vitamin C were thus reflected by the amount of vitamin C present in drinking water as described before [19].

The brain cortex, the heart, the liver, and the spleen were harvested from the exact same mice (for which we had serum vitamin C measurements) to identify and quantify proteins by label-free LC-MS/MS (study design in Fig 1B). We aimed to uncover proteins that significantly correlated with serum vitamin C levels in each of these organs. Overall, 3019, 1891, 2900, and 3592 proteins were identified in the brain, the heart, the liver, and the spleen of all the female groups, respectively (detailed in S2 Table). For the male groups, 2778, 1929, 2821, and 3822 proteins were identified in the brain, the heart, the liver, and the spleen, respectively (detailed S3 Table). Normalization of the data from all the organs under study was performed on the LFQ intensities using MaxQuant. As indicated in S1 Fig, the distribution of the log2 transformed protein intensities were comparable as the data for each sample were distributed at similar levels for either the brain, the heart, the liver, or the spleen in both females and males.

To examine the variation among the different biological replicates within the same experimental groups for each organ, multi-scatter plots were generated using Perseus software. More specifically, Pearson correlation coefficients between the different triplicates within the various treatment groups (GL00, GL01, GL05, GL40, GLR40, and WT00 for females or males) varied from 0.9156 to 0.9894 for the brain (S2 Fig). The Pearson correlation coefficients varied from 0.8810 to 0.9871 for the heart (S3 Fig). The Pearson correlation coefficients varied from 0.8668 to 0.9914 for the liver (S4 Fig). The Pearson correlation coefficients varied from 0.9170 to 0.9839 for the spleen (S5 Fig). These results indicated a high degree of correlation among different triplicates within each treatment groups for the different organs. Altogether, the LFQ

intensity distribution and the Pearson correlation analyses indicated that the LC-MS/MS experiments were reproducible between biological replicates for all organs.

## Commonality and tissue specificity of organ proteomes

A previous study on hepatic microsomal enriched fractions of *Gulo*$^{-/-}$ mice revealed differences between the proteome profiles of females and males [18]. Hence, females and males LC-MS/MS data were analyzed independently. As shown in Fig 2A, 1035 proteins were identified in all four organs of the female mice. In contrast, 817, 203, 705, and 1029 proteins were distinctive to brain, heart, liver, and spleen, respectively. In the males, 1031 proteins were identified in all four organs (Fig 2B). In contrast, 693, 229, 614, and 1198 proteins were distinctive to brain, heart, liver, and spleen, respectively, in the males. Lists of proteins common and specific to the four organs under study are presented in S4 Table for females and males. Biological processes associated with each of these lists of proteins were evaluated using the Database for Annotation, Visualization, and Integration Discovery (DAVID) tool [23]. The gene ontology (GO) terms are presented in the S5 and S6 Tables for females and males, respectively. The biological

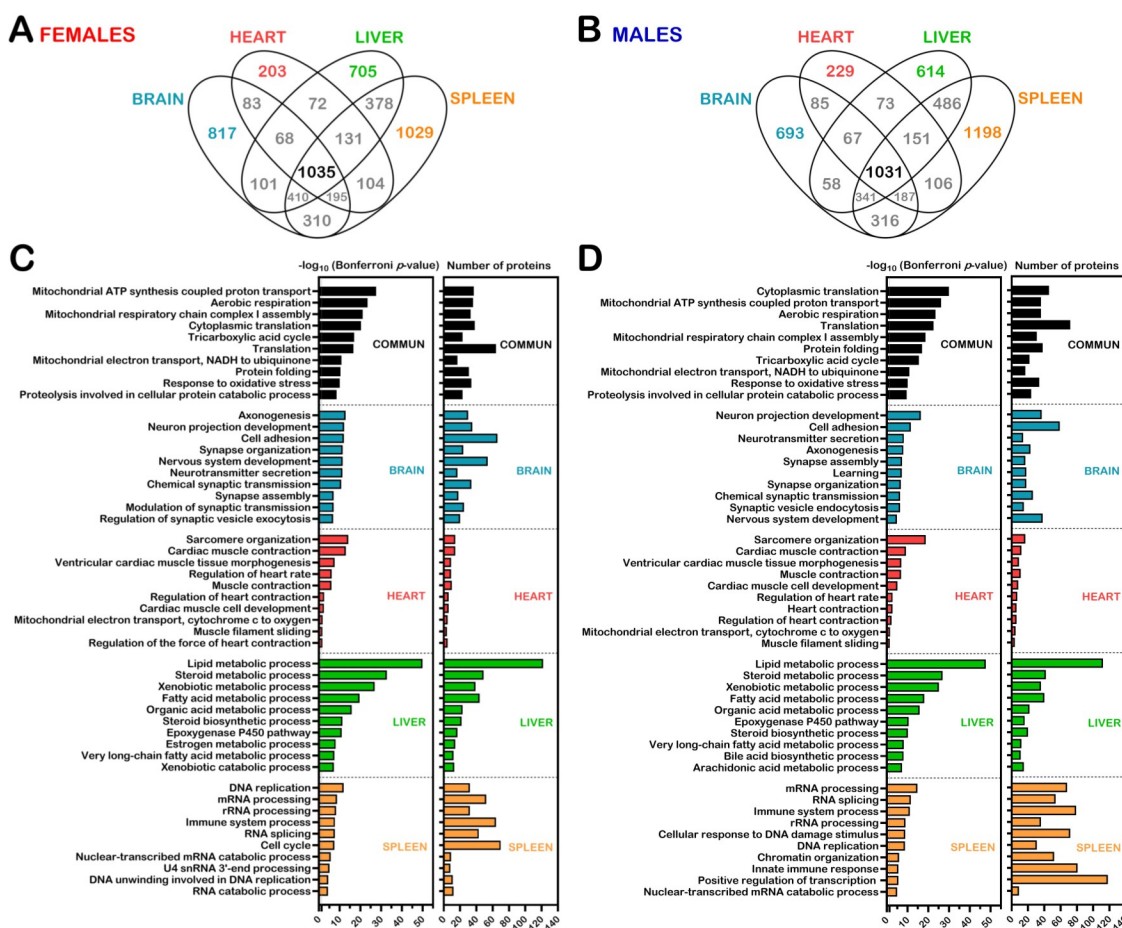

**Fig 2. Commonality and tissue specificity of organ proteomes.** (A) Venn diagram showing the number of identified proteins in all and in specific organs of the female cohort. (B) Venn diagram showing the number of identified proteins in all and in specific organs of the male cohort. (C) Biological processes identified in specific organs and in all four organs analyzed in females. (D) Biological processes identified in specific organs and in all four organs analyzed in males. For C and D, the graph on the left shows the top ten identified gene ontology term with a Bonferroni *p*-value < 0.05 and the graph on the right shows the number of proteins characterizing each biological process from the proteomic data.

processes that were common between females and males are highlighted in yellow in these supplementary Tables. Fig 2C and 2D present the top ten GO terms for females and males in each organ, respectively. The top GO terms corresponding to proteins identified in all four organs in both females and males coincided to processes involved in mitochondrial energy production, translation, and protein homeostasis. The GO terms specific to brain proteins included different aspects of axon or synapse organization and nervous system development. The GO terms distinctive to heart proteins included sarcomere organization, cardiac muscle morphogenesis and contraction. The GO terms specific to liver proteins coincided to the epoxygenase P450 pathway and lipid, steroid, xenobiotic, organic acid, and cholesterol metabolisms. Finally, the GO terms distinctive to spleen proteins included DNA replication, mRNA and rRNA processing, and processes of the immune system.

## Vitamin C levels differentially impact the proteome profiles of various tissues in females and males

We performed principal component analysis (PCA) on the different groups of mice using LFQ normalized and imputed data to uncover sex-based difference in the proteomic profiles of each organ. The different treatment groups were separated to better visualize any sexual dimorphism on each PCA graph. The Fig 3 shows an obvious sexual dimorphism (no overlap between the positions or clusters of the males and females in the individual graphs) in: 1) the brain, heart, liver, and spleen of wild type animals; 2) in the heart, the liver, and spleen of GL40 treated mice; 3) in the brain and liver of GLR40 mice; 4) in the heart, liver, and spleen of GL05 animals; 5) in the liver and spleen of GL01 mice; and 6) in the brain, liver, and spleen of GL00 vitamin C deficient mice. Overall, these results indicated that the proteome profiles generally differed between females and males in the brain, heart, liver, and spleen tissues of mice treated with various concentrations of vitamin C in drinking water.

We performed additional PCAs to visualize the impact of vitamin C deficiency and subsequent vitamin C rescue treatments on the proteome profiles of the different organs under study in both *Gulo*[-/-] females and males. More precisely, we compared the proteome profiles of vitamin C deficient *Gulo*[-/-] mice (FGL00 and MGL00) to *Gulo*[-/-] mice that never experienced a vitamin C deficiency (FGL40 and MGL40) and vitamin C deficient mice that were treated (or rescued) with vitamin C for one month until they reached the age of four months (FGLR40 and MGLR40 groups depicted in Fig 1A). As indicated in Fig 4, comparisons of the proteome profiles of FGL40 and FGLR40 female clusters overlapped each another but did not intersect the FGL00 female cluster in the brain, heart, liver, and spleen PCA graphs. These results indicated that the brain, heart, liver, and spleen proteome profiles of vitamin C deficient *Gulo*[-/-] females were rescued by a one-month vitamin C treatment. A rescue was also observed for males in the brain, liver, and spleen. Indeed, the proteome profiles of MGLR40 individuals clustered with the MGL40 males but did not overlap with the MGL00 males (Fig 4). Finally, the MGL00 male cluster overlapped with both the MGL40 and MGLR40 individuals in the heart PCA graph showing little distinctive differences between the heart proteome profiles in males. Altogether, these results indicated that the impact of vitamin C deficiency and rescue on the proteome profiles of females and males differed in the heart.

## Identification of proteins for which LFQ intensities showed significant correlations with serum vitamin C levels in female and male cohorts for each individual organ

Statistical associations between LFQ intensities of the quantified proteins in each organ and serum vitamin C levels were determined by Spearman correlation analyses with a specific set

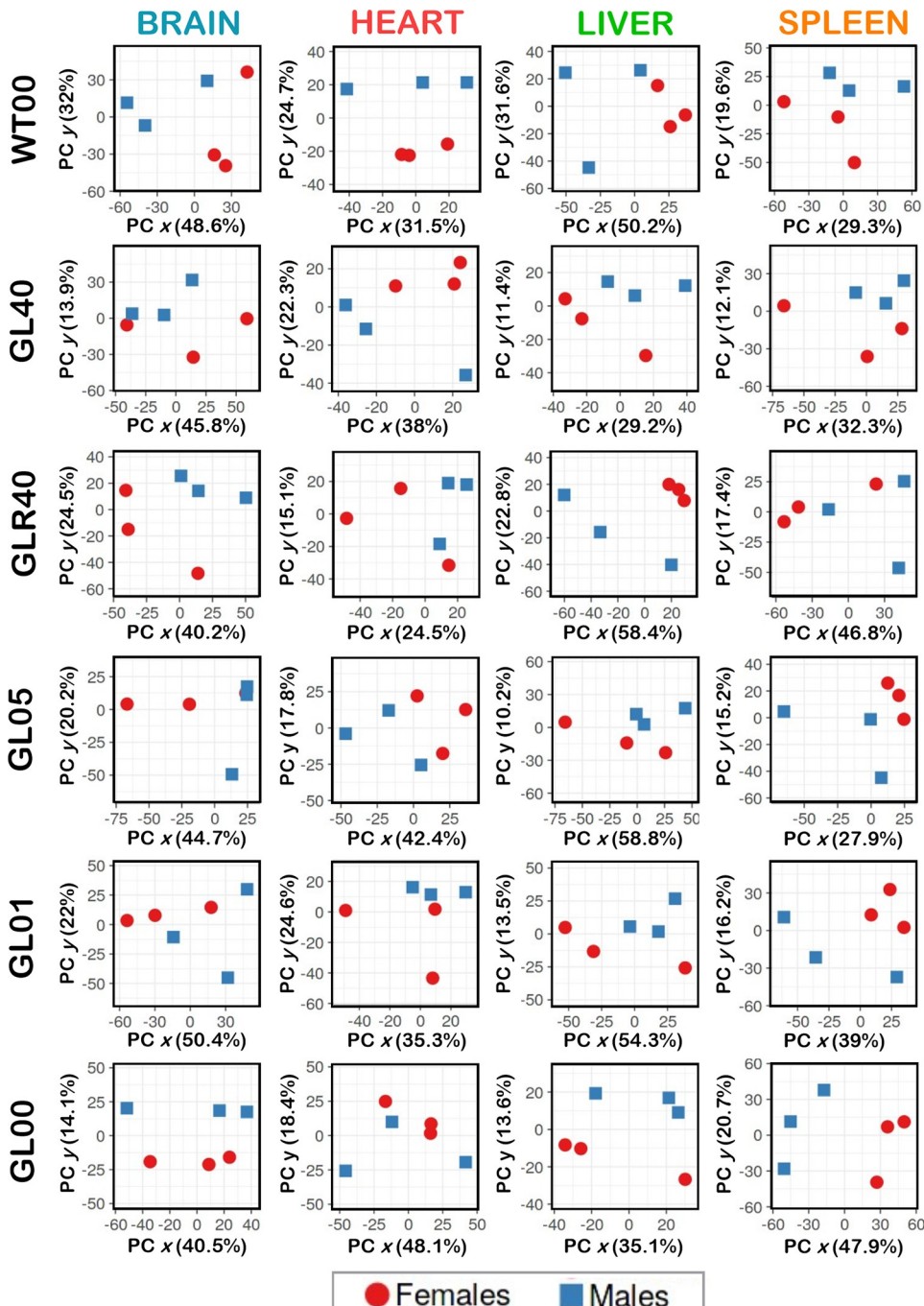

**Fig 3. Principal component analysis (PCA) showing the sexual dimorphism in each experimental mouse group for brain, heart, liver, and spleen tissues.** F = Females and M = Males; WT00 = WT females or males with no ascorbate for 4 months; GL40 = *Gulo*$^{-/-}$ females or males treated with 0.4% ascorbate for 4 months; GLR40 = *Gulo*$^{-/-}$ females or males treated with 0% ascorbate for 1 month followed by 1 month of 0.4% ascorbate treatment; GL05 = *Gulo*$^{-/-}$ females or males treated with 0.05% ascorbate for 4 months; GL01 = *Gulo*$^{-/-}$ females or males treated with 0.01% ascorbate for 4 months; GL00 = *Gulo*$^{-/-}$ females or males with no ascorbate for 1 month; N = 3 for each experimental group.

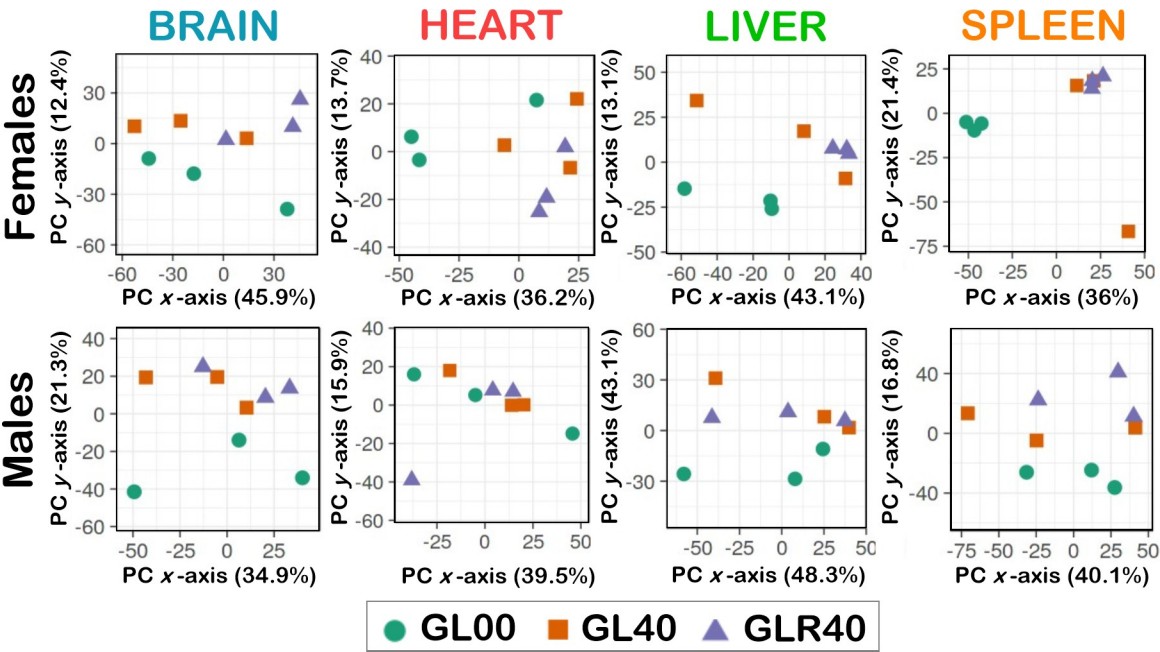

**Fig 4. Principal component analysis (PCA) on the proteomic data in each organ for the vitamin C rescue experiments.** GL00 = *Gulo⁻/⁻* females and males with no ascorbate for 1 month (green circles); GL40 = *Gulo⁻/⁻* females and males treated with 0.4% ascorbate for 4 months (orange squares); GLR40 = *Gulo⁻/⁻* females and males treated with 0% ascorbate for 1 month followed by 1 month of 0.4% ascorbate treatment (purple triangles); N = 3 for each experimental group.

of criteria. First, a Spearman correlation was considered significant with a *p*-value < 0.05 for N = 18 females or males. Secondly, proteins with at least a 1.5-fold change between the vitamin C deficient mice (GL00) and 0.4% vitamin C treated *Gulo⁻/⁻* mice since weaning (GL40) were selected to identify the biological processes affected by vitamin C in the different organs. Based on these criteria, the Fig 5 shows that 24 (0.79% of total quantified proteins) and 33 (1.09%) proteins correlated positively and negatively with serum vitamin C levels in the brain of females, respectively. In the males, 63 (2.27%) and 51 (1.84%) proteins correlated positively and negatively with serum vitamin C levels in the brain, respectively. The scatter plots of the heart indicated that 46 (2.43%) and 83 (4.39%) proteins correlated positively and negatively with serum vitamin C, respectively, in the females. In the males, 21 (1.09%) proteins correlated positively and negatively with serum vitamin C levels. For the liver, the Fig 5 shows that 242 (8.34%) and 111 (3.83%) proteins correlated positively and negatively with serum vitamin C levels in females, respectively. In the males, 36 (1.28%) and 129 (4.57%) proteins correlated positively and negatively with serum vitamin C in the liver, respectively. Finally, 303 (8.44%) and 324 (9.02%) proteins correlated positively and negatively with serum vitamin C levels in the spleen of females. In the males, 32 (0.84%) and 35 (0.92%) proteins correlated positively and negatively with serum vitamin C levels, respectively. Overall, there were more identified proteins that correlated with serum vitamin C levels (positively and negatively) in the heart, liver, and spleen of females than in males (Fig 5). In contrast, there were more proteins correlating with serum vitamin C in the brain of males than in females. The lists of proteins for which LFQ intensities correlated with serum vitamin C levels in the four different organs are shown in S7 and S8 Tables for the females and males, respectively.

Biological processes associated with the lists of proteins correlating with serum vitamin C in each organ were evaluated using the DAVID bioinformatics tool [23]. The identity of the proteins associated with GO terms significantly associated with serum vitamin C levels are

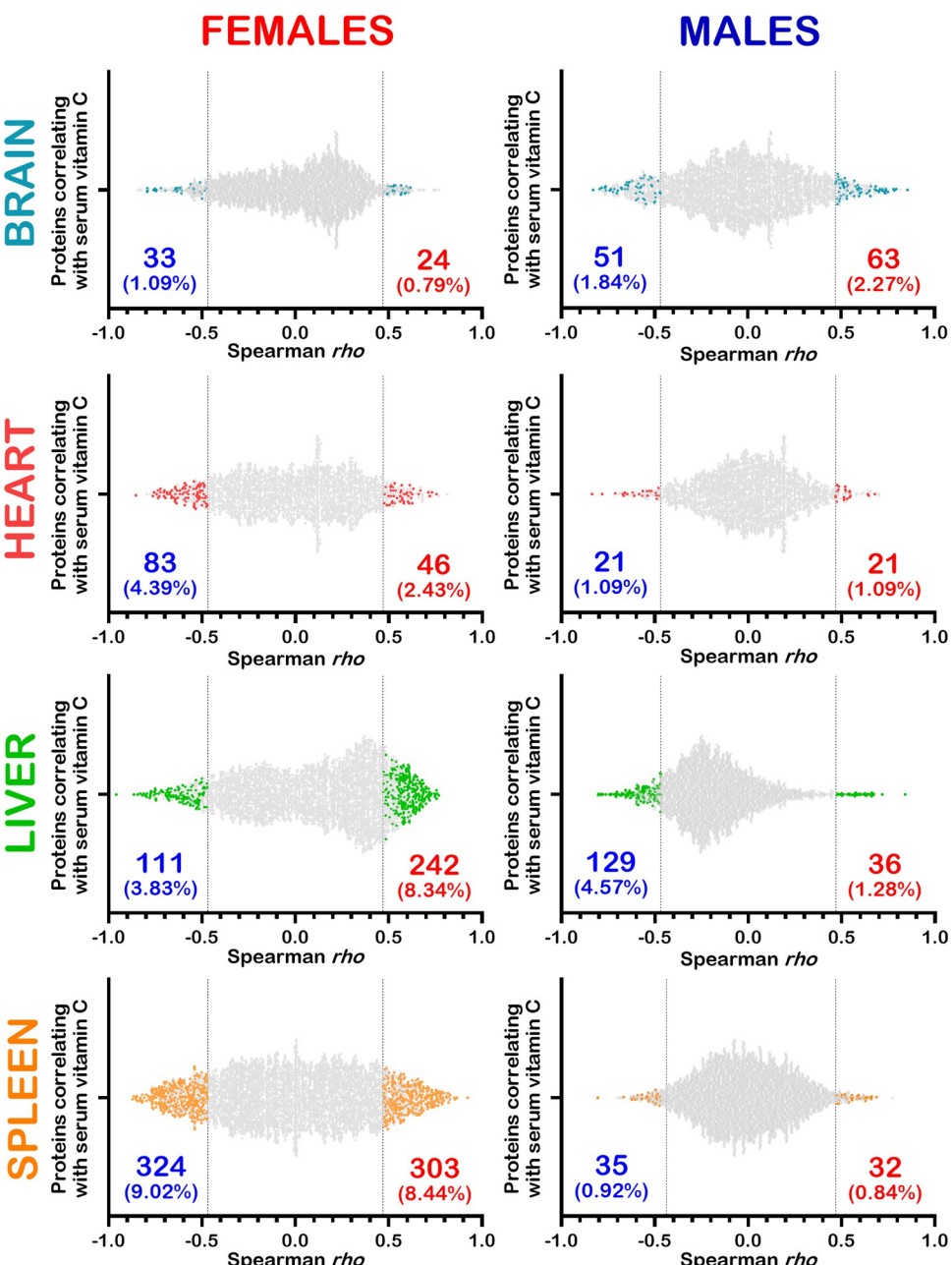

**Fig 5. Scatter plots showing the number of proteins correlating significantly with serum vitamin C levels in different organs of *Gulo*⁻/⁻ females and males.** Correlations were considered significant for proteins with at least a 1.5-fold change between the vitamin C deficient mice (GL00) and 0.4% vitamin C treated *Gulo*⁻/⁻ mice since weaning (GL40) if the Spearman correlation *p*-value was less than 0.05 (N = 18 females or males). The percentage of proteins correlating negatively (in blue) and positively (in red) are also indicated on each graph. The total number of quantifiable proteins for the brain, the heart, the liver, and the spleen were 3019, 1891, 2900, and 3592 proteins respectively for the females and 2778, 1929, 2821, and 3822 proteins respectively for the males.

indicated S9 Table for each organ in females and males. As shown in Fig 6, several subunits of the mitochondrial respiratory chain complex I correlated inversely with serum vitamin C levels in male brain tissues. In addition, sets of proteins involved in glycolytic and lipid metabolic processes correlated positively with serum vitamin C levels in the brain tissue of males. The

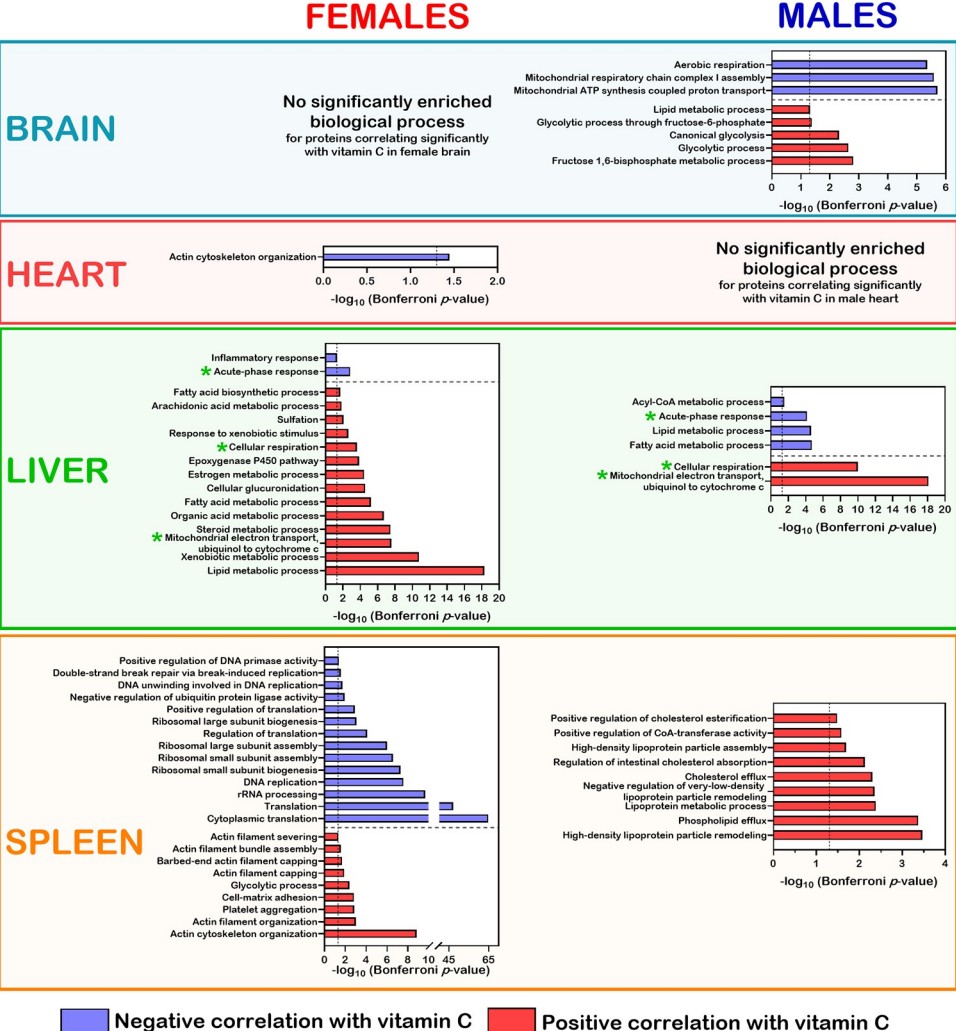

**Fig 6. Biological processes significantly represented (with a Bonferroni *p*-value < 0.05) by proteins correlating inversely or positively with serum vitamin C levels in each organ of female and male cohorts.** The green asterisks represent biological processes that were commonly affected in the liver of both females and males.

data in S9 Table indicated that all the proteins composing these biological processes were not brain specific as these proteins were also identified in the other organs under study. The proteins in the brain of females that significantly correlated with serum vitamin C levels did not show enrichment for specific GO terms after Bonferroni adjustment (detailed in S9 Table).

The GO analysis of the heart tissue revealed that proteins involved in actin cytoskeleton organization correlated inversely with serum vitamin C in the females (Fig 6). No GO term was identified for proteins correlating positively with serum vitamin C after a Bonferroni adjustment. The S9 Table indicated that the Xin actin binding repeat containing 1 (Xirp1) protein was the only heart muscle specific factor correlating inversely with serum vitamin C that was part of the actin cytoskeleton organization in females. The other proteins of this biological process were also identified in the other organs under study, even though actin cytoskeleton organization did not correlate negatively with vitamin C in these other organs. The proteins in the heart of males that significantly correlated with serum vitamin C levels did not show enrichment for specific GO terms after Bonferroni adjustment (detailed in S9 Table).

The GO analysis of the liver tissue indicated that more biological processes were affected by serum vitamin C alterations compared to the brain and the heart tissues. Proteins of the acute phase and inflammatory response were inversely correlated with serum vitamin C levels in the liver of females. In addition to acute phase response, male liver tissues also exhibited a negative correlation of proteins involved in fatty acid, lipid, and acyl-CoA metabolic processes (Fig 6). Although the female liver tissues showed a positive correlation of lipid and fatty acid metabolic processes with serum vitamin C levels, the list of proteins associated with such biological processes were different from the list of proteins found in the males (S9 Table). Other biological processes that correlated positively with serum vitamin C in the liver of females included xenobiotic, steroid, organic acid, estrogen, and arachidonic acid metabolic processes. Based on the S9 Table, 30% and 61% of the proteins related to the biological processes correlating negatively or positively with serum vitamin C in males and females, respectively, were not identified in the brain, heart, or spleen by LC-MS/MS in the present study. Thus, the vitamin C deficiency affected several biological processes that, by in large, are at least more abundant in the liver tissue. In contrast, proteins associated with the mitochondrial electron transport (ubiquinol to cytochrome c) and cellular respiration correlated positively with serum vitamin C levels in both females and males (detailed in S9 Table) as it has been described before for the liver of *Gulo*<sup>-/-</sup> mice [18]. Although proteins of the mitochondrial electron transport system or involved in cellular respiration were also identified in all the organs under study, these biological processes did not correlate with serum vitamin C in the brain, heart, or spleen.

The GO analysis of the spleen tissue revealed that more biological processes were significantly correlating with serum vitamin C in the females than in the males. Proteins mainly involved in various aspects of cytoplasmic translation, ribosomal biogenesis, and DNA replication were inversely correlating with serum vitamin C levels in the spleen of females. No significant biological process was correlating negatively with serum vitamin C in the spleen of males. The proteome of female spleen also exhibited a positive correlation of proteins associated with different aspects of actin cytoskeleton organization, platelet aggregation, and glycolytic processes with serum vitamin C levels (Fig 6). In contrast, the proteome of male spleen tissues exhibited a positive correlation of proteins involved in different aspects of lipoprotein particle remodeling and phospholipid/cholesterol efflux with serum vitamin C levels. Based on the S9 Table, 29% of the proteins composing the biological processes correlating (positively or negatively) with serum vitamin C levels in the females were identified by LC-MS/MS only in the spleen tissue in the current study. All the proteins associated with the different aspects of lipoprotein particle remodeling and phospholipid/cholesterol efflux in the male spleen were also identified and quantified in the brain, heart, and spleen by LC-MS/MS. However, such biological processes did not correlate with vitamin C levels in these other organs.

Altogether, these observations indicated that proteins involved in different biological processes were correlating with serum vitamin C levels in the brain, heart, liver, and spleen. The acute phase response, the ubiquinol to cytochrome c mitochondrial electron transport, and cellular respiration were the only biological processes that were similarly affected in the liver of both females and males (Fig 6).

## Identification of proteins for which LFQ intensities showed significant correlations with serum vitamin C levels in at least three organs in females and/or males

As indicated above, different biological processes were affected by vitamin C deficiency in the various organs of both females and males. Nevertheless, several proteins that are not significantly associated with specific gene ontology terms could correlate with serum vitamin C levels

in several tissues. We thus determined which specific genes exhibited similar protein level alterations in several organs of *Gulo*$^{-/-}$ mice treated with different levels of vitamin C in drinking water. The Venn diagrams in Fig 7A show the number of proteins that correlated significantly with serum vitamin C concentrations in the various organs under study (with a Spearman *p*-value < 0.05 and a 1.5-fold difference between GL00 and GL40) in both females and males. The S10 Table provides the lists of proteins for each section of the Venn diagrams shown in Fig 7A. We focused on the proteins that correlated significantly with serum vitamin C levels in at least three organs (sections highlighted in dark blue or dark red in the Venn diagrams). Overall, 17 and two proteins met these criteria in females and males, respectively. The Fig 7B presents heatmaps of all these proteins in both females and males. The color scale at the bottom of the heatmaps represents the Spearman *rho* value for each protein in each indicated organ. The blue and red rectangles in the heatmaps indicate negative and positive correlations, respectively. The darker the color, the higher the absolute *rho* values. The "+" signs indicate significant correlations (*p*-value < 0.05) with at least 1.5-fold difference between the GL00 and GL40 mouse groups.

A significant inverse correlation with serum vitamin C levels was observed for the Igkc, Lrg1, Rbm3, Ighg2b proteins in all the four organs under study in females (heatmap on the left in Fig 7B). Although not significant in all organs under study, a similar inverse correlation pattern was observed for Lrg1 and Rbm3 proteins in males. Significant inverse correlations with serum vitamin C levels were observed for the Fetub, Hpx, S100a8, S100a9, Cpq, Igkv14-111, Cfi, Ighg2c, and Col1a1 proteins in at least three of the four organs under study (heatmap on the left in Fig 7B). The correlation patterns for these proteins were different in the male tissues (except for Cpq and Cfi) often with Spearman *p*-values not significant (heatmap on the right in Fig 7B). Significant positive correlations with serum vitamin C levels were observed for the Ftl1, Fth1, As3mt, and Apoc3 proteins in at least three of the organs under study in females (heatmap on the left in Fig 7B). Although not always significant, the correlations for these proteins in the same tissues correlated positively with serum vitamin C levels in the males (heatmap on the right in Fig 7B). The C3 and Serpina3k proteins correlated inversely and positively with serum vitamin C levels, respectively, in at least three organs in the males. The females did not show significant correlation with serum vitamin C in any of the organs in the present study (bottom panels in Fig 7B).

Finally, gene ontology analysis indicated that the proteins correlating inversely with serum vitamin C levels in most organs in females were associated with the innate immune response or inflammation (with a Bonferroni adjustment *p*-value < 0.0053). Proteins involved in inflammatory response included Igkc, Lrg1, Ighg2b, Fetub, Hpx, S100a8, S100a9, Igkv14-111, Cfi, Ighg2c, and C3. Thus, a vitamin C deficiency induces an acute inflammatory response in the organs under study, especially in the females.

## Integration of the serum proteome profile

To determine whether the proteins correlating with serum vitamin C levels in several organs of our mouse cohorts could also be detected in the serum, we revisited the mass spectrometry data previously published on the serum proteome profile of the exact same mice treated with different concentrations of vitamin C in drinking water [19]. Statistical associations between LFQ intensities of the quantified serum proteins and serum vitamin C levels were determined by Spearman correlation analyses (*p*-value < 0.05 for N = 18 females or males and with at least a 1.5-fold change between the vitamin C deficient mice (GL00) and 0.4% vitamin C treated *Gulo*$^{-/-}$ mice since weaning). The scatter plot in Fig 8A indicates that 62 and 63 proteins correlated negatively and positively with serum vitamin C levels in females, respectively. In males,

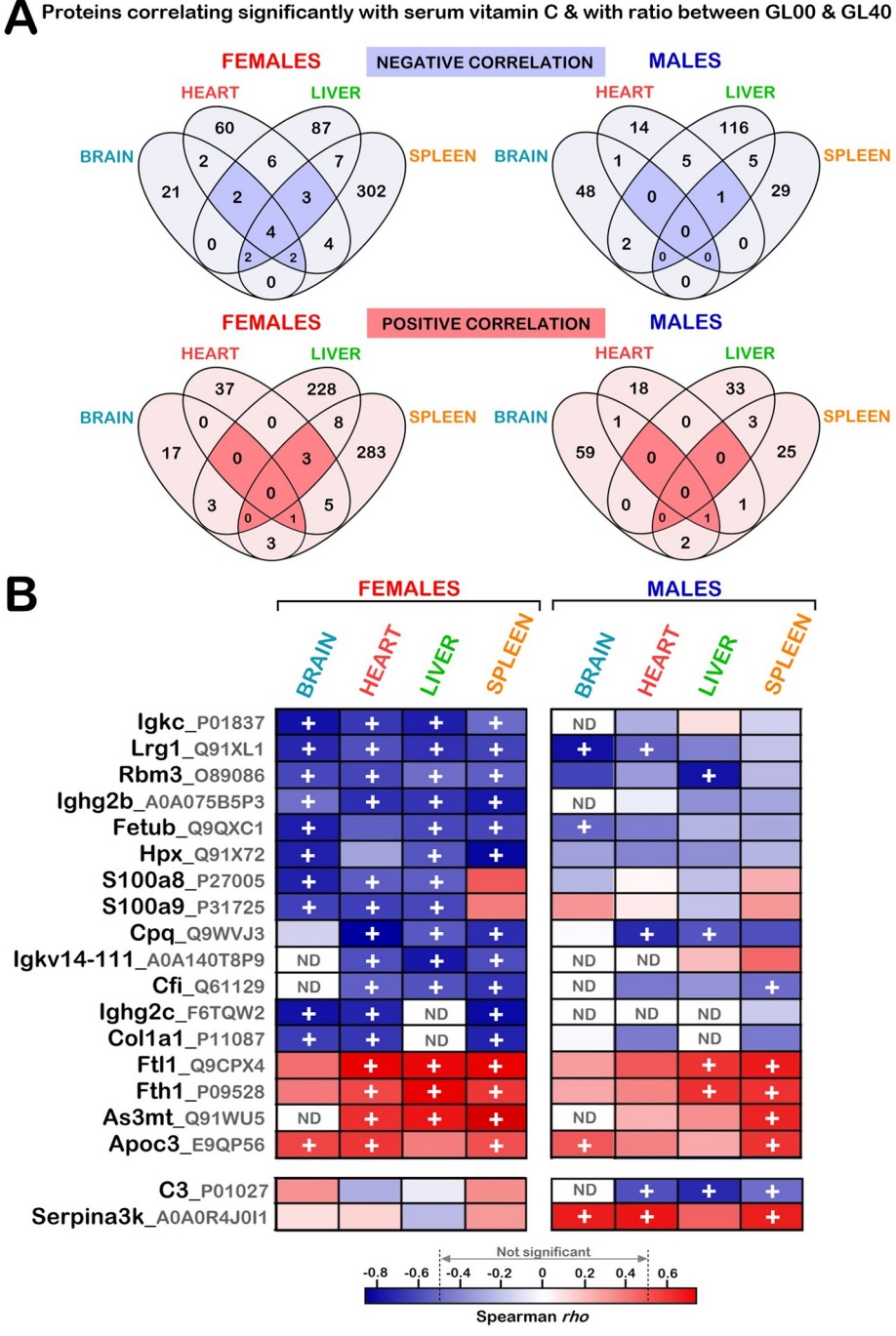

**Fig 7. Proteins correlating significantly with serum vitamin C in more than three organs.** (A) Venn diagram presenting the number of proteins correlating with serum vitamin C in one or more than one organ in females and males. The numbers highlighted in blue or red in the Venn diagrams represent the proteins correlating with serum vitamin C in at least three organs (with a Spearman correlation $p$-value < 0.05 and a difference of at least 1.5-fold between GL00 and GL40 mice). (B) Heatmap of the proteins correlating significantly with serum vitamin C levels in at least three organs in females (top panels) and males (bottom panels). Negative and positive Spearman correlation $rho$ values are in blue and red, respectively. The "+" symbol represents proteins with at least a 1.5-fold difference and a Spearman correlation $p$-value < 0.05 between GL00 and GL40 groups in the organs under study. ND = protein not detected in the indicated organ samples.

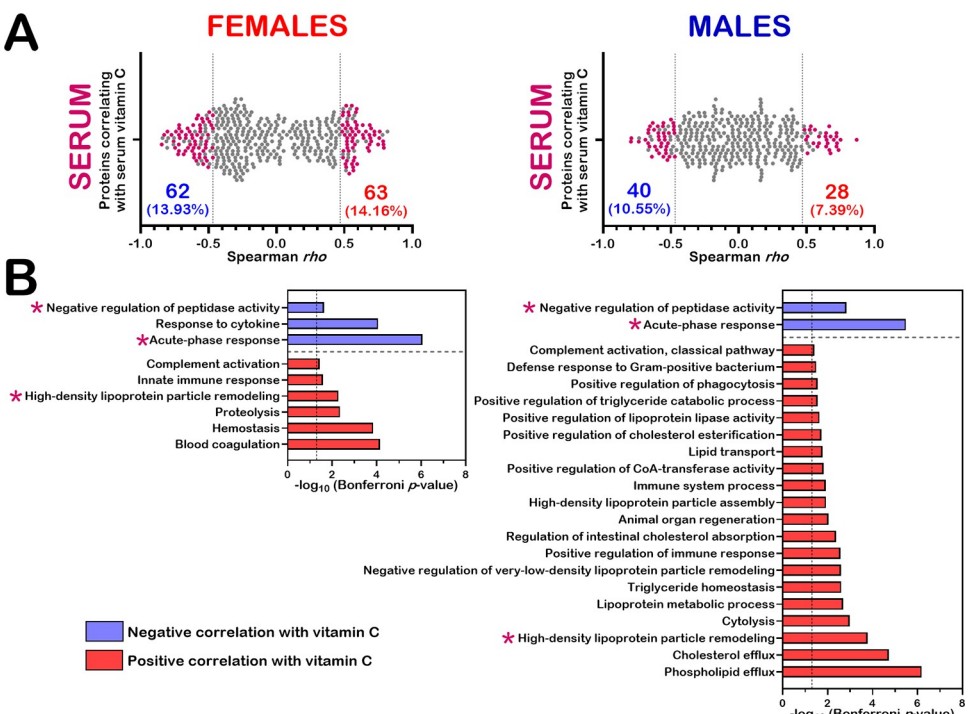

**Fig 8. Serum proteins correlating significantly with serum vitamin C levels.** (A) Scatter plots showing the number of proteins correlating significantly with serum vitamin C levels in $Gulo^{-/-}$ females and males (with a Spearman correlation $p$-value $< 0.05$ and a difference of at least 1.5-fold between GL00 and GL40 mice). The percentage of proteins correlating negatively (in blue) and positively (in red) are also indicated on each graph. (B) Biological processes significantly represented (with a Bonferroni $p$-value $< 0.05$) by proteins correlating inversely or positively with serum vitamin C levels in females and males. The asterisks in purple represent the biological processes common in both females and males.

40 and 28 proteins correlated negatively and positively with serum vitamin C levels, respectively. The lists of these proteins are shown in the S11 Table. Biological processes associated with the lists of proteins correlating with serum vitamin C were also evaluated using DAVID [23]. The identity of the proteins associated with GO terms significantly associated with serum vitamin C levels in females and males are indicated S12 Table. As shown in Fig 8B, more biological processes were affected by ascorbate levels in the serum of males (22 functional modules) than in females (9 modules). Biological processes common in both females and males included the negative regulation of peptidase activity, the acute-phase response (for negative correlations with vitamin C), and high-density lipoprotein particle remodeling (for positive correlation with vitamin C).

The Fig 9 shows five proteins that were detected in the serum and that were significantly correlating with serum vitamin C levels in at least three organs in females. Such proteins included Lrg1, Col1a1, S100a9, Ftl1, and Cpq. Interestingly, serum Lrg1 levels correlated inversely with serum vitamin C levels like in the brain, heart, liver, and spleen tissues. Negative correlations for Lrg1 were obtained in the serum and the organs of males (Fig 9). The detection of Col1a1 in the serum also negatively correlated with serum vitamin C levels in the females. However, mass spectrometry did not detect Col1a1 proteins in the liver of females (or males). The S100a9 protein levels correlated inversely with vitamin C in the serum, brain, heart, and liver of females (Fig 9). However, S100a9 correlated positively with serum vitamin C in the

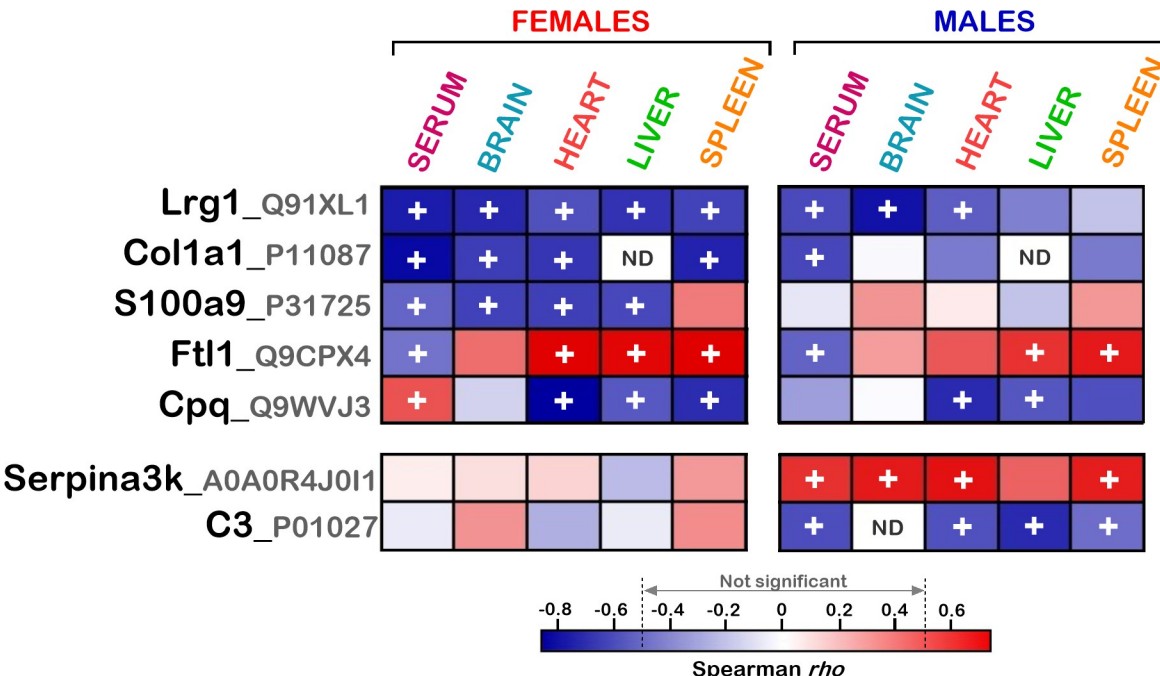

**Fig 9. Proteins correlating significantly with serum vitamin C levels in more than three organs and in the serum.** Heatmap of the proteins correlating with serum vitamin C levels in at least three organs in females (top panels) and males (bottom panels). Negative and positive Spearman correlation *rho* values are in blue and red, respectively. The "+" symbol represents proteins with at least a 1.5-fold difference and a Spearman correlation *p*-value < 0.05 between GL00 and GL40 groups in the organs under study. ND = protein not detected in the indicated serum or organ samples.

spleen of females. The correlation pattern of S100a9 with serum vitamin C in the various tissues differed in the males compared to females. Ftl1 levels correlated positively with serum vitamin C in all four organs under study (Fig 9) in both females and males. In contrast, serum Ftl1 levels correlated inversely with serum vitamin C levels in both females and males. The Cpq protein level in the serum correlated positively with serum vitamin C but its level in the heart, liver, and spleen correlated inversely with serum vitamin C levels in the females (Fig 9). Serum Cpq levels in the males did not significantly correlate with serum vitamin C levels (negative tendency in Fig 9).

Two proteins were detected in the serum and were significantly correlating with serum vitamin C levels in at least three organs in males. The abundance of the Serpina3k protein in the serum, brain, heart, liver, and spleen correlated positively with serum vitamin C levels (Fig 9). The correlations of Serpina3k protein with serum vitamin C levels in the different organs of females were not statistically significant. The complement factor C3 correlated inversely with serum vitamin C in the heart, liver, spleen, and serum of males (Fig 9). The correlations of C3 with serum vitamin C were different in the females and non-significant.

## Validation of the differential abundance of Ftl1, S100a9, Uqcrc1, and Uqcrfs1 proteins by immunoblot analyses in the different organs

Ftl1 and S100a9 proteins were further analysed by western blots on total lysates from brain, heart, liver, and spleen of vitamin C deficient *Gulo*$^{-/-}$ mice (GL00) and 0.4% vitamin C treated *Gulo*$^{-/-}$ mice since weaning (Fig 10). Quantifications of each protein in the immunoblots are presented in Fig 11. The western experiments indicated that the abundance of Ftl1 protein was significantly decreased in the brain, liver, and spleen of vitamin C deficient females (FGL00)

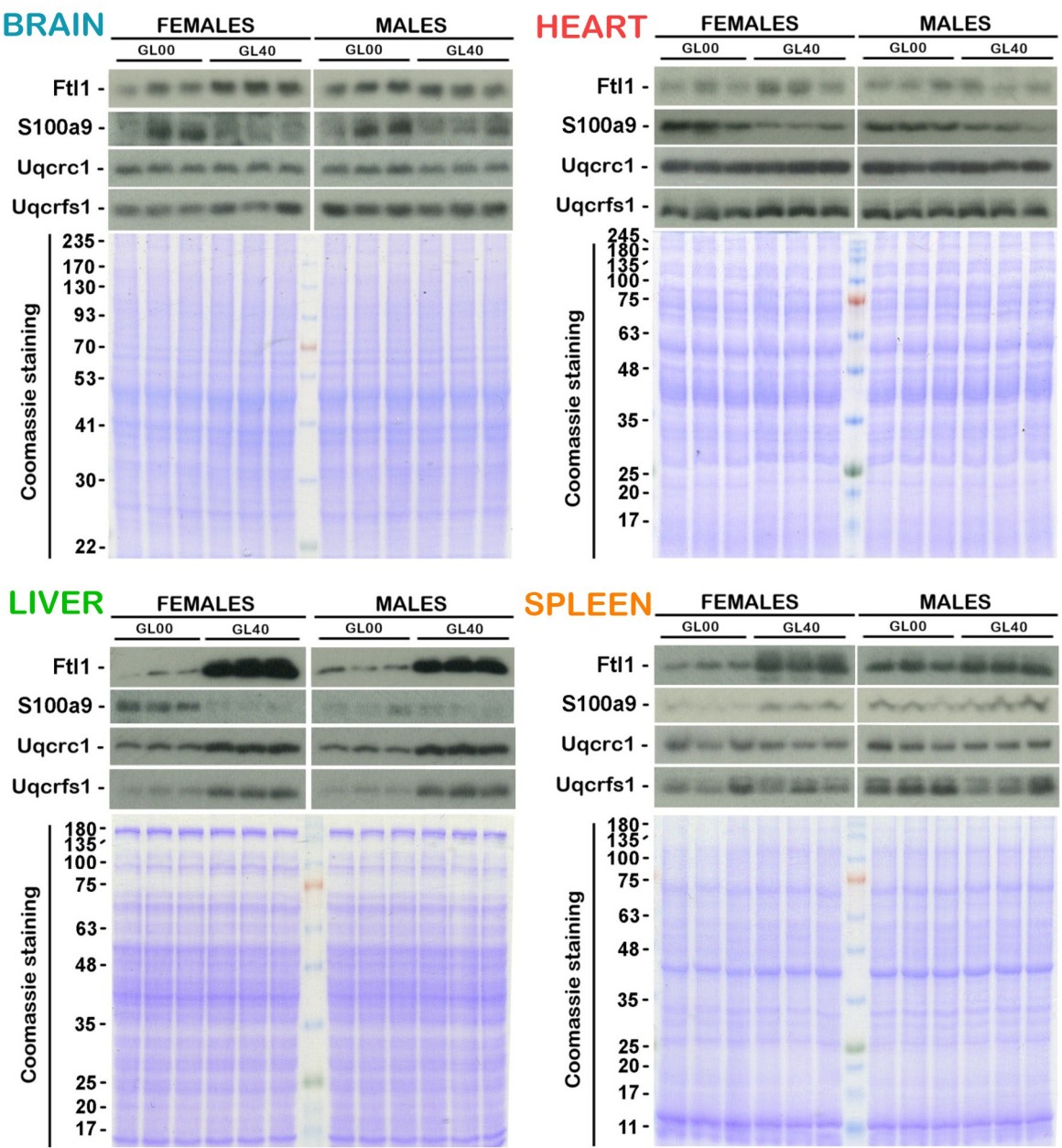

**Fig 10. Immunoblotting analysis of Ftl1, S100a9, Uqcrc1, and Uqcrfs1 on brain, heart, liver, and spleen samples.** Analyses of four different proteins by western blots on the whole brain, heart, liver, and spleen lysates for vitamin C deficient mice (GL00) and 0.4% vitamin C treated *Gulo*$^{-/-}$ mice since weaning (GL40). Coomassie staining was used as in-gel loading control. (N = 3 females and 3 males for each cohort).

compared to 0.4% vitamin C treated *Gulo*$^{-/-}$ females (FGL40) (Fig 11; Student *t*-test *p*-value < 0.05). A decreased tendency was observed in the heart of vitamin C deficient females. The abundance of Ftl1 protein was significantly decreased only in the liver and spleen of vitamin C deficient males (Fig 10). Such results are consistent with the LFQ data presented as histograms in S6 Fig for the same proteins and the same groups (GL00 versus GL40 for females and males).

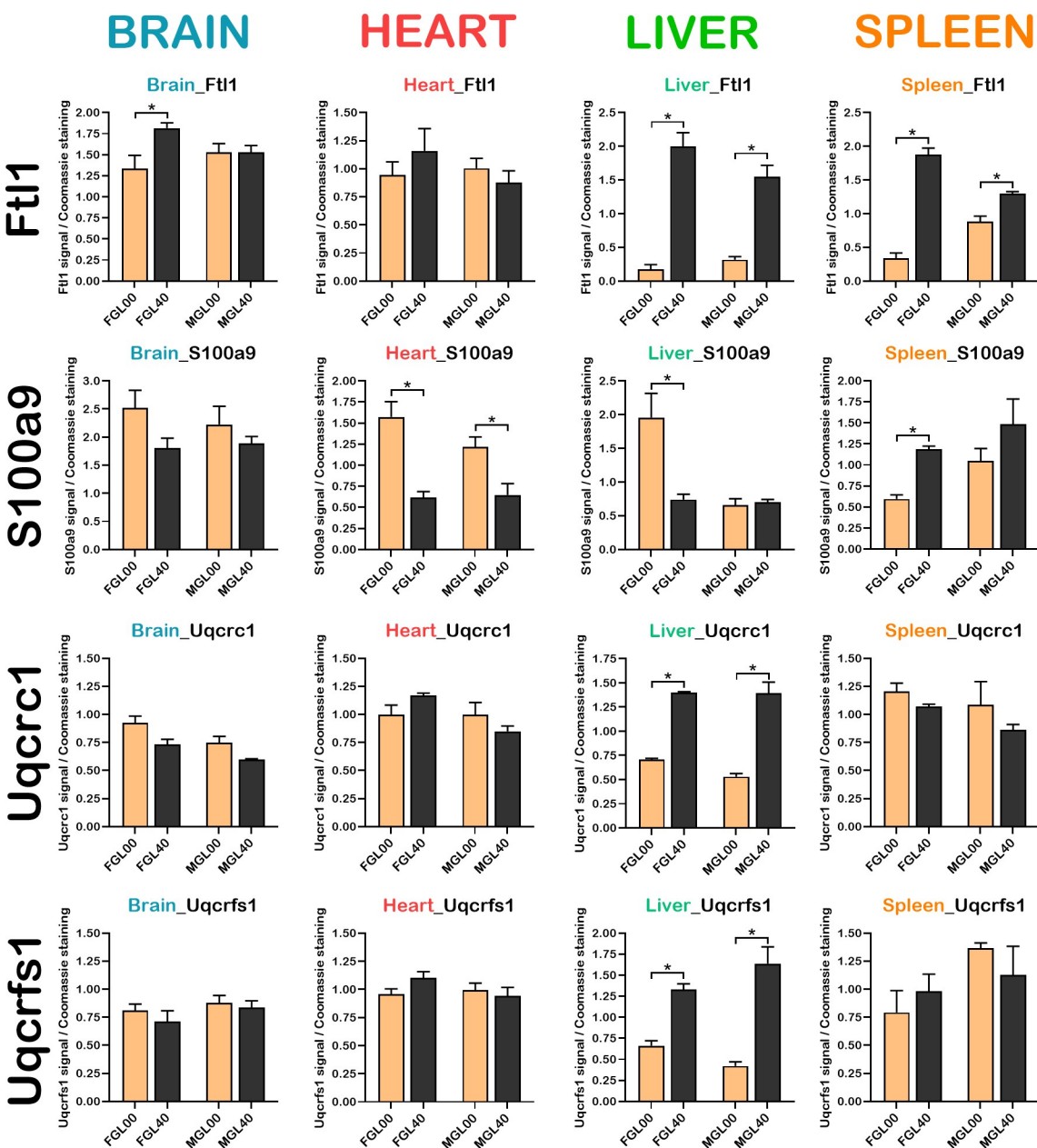

**Fig 11. Signal quantification of Ftl1, S100a9, Uqcrc1, and Uqcrfs1 proteins from immunoblotting analysis.** Protein signal depicted on the western blots of the Fig 10 were quantified. Histograms present the mean of the signal quantification for Ftl1, S100a9, Uqcrc1, and Uqcrfs1 over Coomassie staining signal. Bars present the standard error of the mean. (N = 3 females and 3 males for each cohort; * $p$-value < 0.05; Welch's $t$-test).

The abundance of S100a9 protein was significantly increased in the heart and liver but significantly decreased in the spleen of vitamin C deficient females (FGL00) compared to 0.4% vitamin C treated *Gulo*$^{-/-}$ females (FGL40) (Figs 10 and 11). This pattern of protein abundance observed by western analyses was similar to the LFQ data for the *Gulo*$^{-/-}$ females (S6 Fig). The level of S100a9 protein was significantly increased in the heart of vitamin C deficient males (MGL00) compared to 0.4% vitamin C treated *Gulo*$^{-/-}$ males (MGL40) (Figs 10 and 11). Although there was a tendency for an increase of S100a9 abundance in 0.4% vitamin C treated

*Gulo*[-/-] males compared to vitamin C deficient males (by ~45% in Fig 11), this tendency was not observed in the LFQ data for the males (S6 Fig).

Since Ftl1 and S100a9 are secreted in the blood, we next examined their levels in the serum of *Gulo*[-/-] mice by western blot analysis. As indicated in the Fig 12A and 12B, the abundance of Ftl1 was significantly increased in the serum of both the *Gulo*[-/-] females and males deficient in vitamin C compared to 0.4% vitamin C treated *Gulo*[-/-] mice. Overall, the results were consistent with the LFQ data (Fig 12C). The abundance of S100a9 was significantly increased (by ~20%) in the serum of vitamin C deficient *Gulo*[-/-] females compared to 0.4% vitamin C treated *Gulo*[-/-] females (Fig 12A and 12B). There was also a tendency for S100a9 to be increased (by ~15%) in the vitamin C deficient *Gulo*[-/-] males. The increase pattern of S100a9 abundance in vitamin C deficient *Gulo*[-/-] mice on the western blots is consistent with the LFQ data (Fig 12C).

The LFQ analyses also indicated that the abundance of the Uqcrc1 and Uqcrfs1 subunits of the mitochondrial complex III was significantly decreased in the liver of vitamin C deficient *Gulo*[-/-] mice (GL00) compared to 0.4% vitamin C treated *Gulo*[-/-] females and males (GL40) but was not significantly changed in the brain, heart, and spleen. Immunoblotting analyses were thus performed with antibodies against Uqcrc1 and Uqcrfs1 proteins (Fig 10). Quantifications of the immunoblots were conducted for each protein and are presented in the graphs of Fig 11. The immunoblots analyses indicated that the abundance of Uqcrc1 and Uqcrfs1 proteins was significantly decreased in the liver of vitamin C deficient *Gulo*[-/-] mice compared to 0.4% vitamin C treated *Gulo*[-/-] mice (Fig 11; in both females and males) confirming the LFQ data (S6 Fig). There was no significant difference in the abundance of Uqcrc1 and Uqcrfs1 proteins in the other organs under study (Figs 10 and 11).

Overall, the results from the western blot analyses were coherent with the LFQ LC-MS/MS data from the tissues and the serum.

## Discussion

In the present study, we performed LC-MS/MS to identify and quantify proteins that correlate with serum vitamin C levels in the brain, heart, liver, and spleen tissues of *Gulo*[-/-] mice. The major aim of the present work was to determine whether similar biological processes were affected by various levels of serum vitamin C in different tissues. The main strength of our study is that serum, brain, heart, liver, and spleen were harvested from the same mice to measure serum vitamin C concentrations and perform mass spectrometry analyses on different tissues for Spearman correlation computation. Venn diagrams on the lists of proteins identified and quantified in the four tissues in females and males not only revealed sets of proteins that were more abundantly identified in specific tissues but also a series of proteins that were common to all the tissues under study. Furthermore, although multi subcellular fractionation experiments were not performed to obtain a deeper proteome coverage, our proteomic results are consistent with previous published reports on the organ specific proteome profiles of various murine tissues [2, 24].

PCA on the different groups of mice indicated a sexual dimorphism regarding the proteome profiles of brain, heart, liver, and spleen tissues. These observations are consistent with previous proteomic reports on murine hepatic tissue and serum samples from *Gulo*[-/-] mice [18, 19]. The observed sexual dimorphic proteome profiles uncovered in the four organs in this study is also in agreement with recent mass spectrometry analyses of extracellular matrix proteins from 25 mouse organs and multiple reaction monitoring assays for protein quantifications in 20 distinct mouse tissues that revealed sex-specific differences in protein abundances in many organs [3, 4]. The reasons for the different proteome profiles between females and males in their various organs are likely due to sexual hormone dimorphism [25–28].

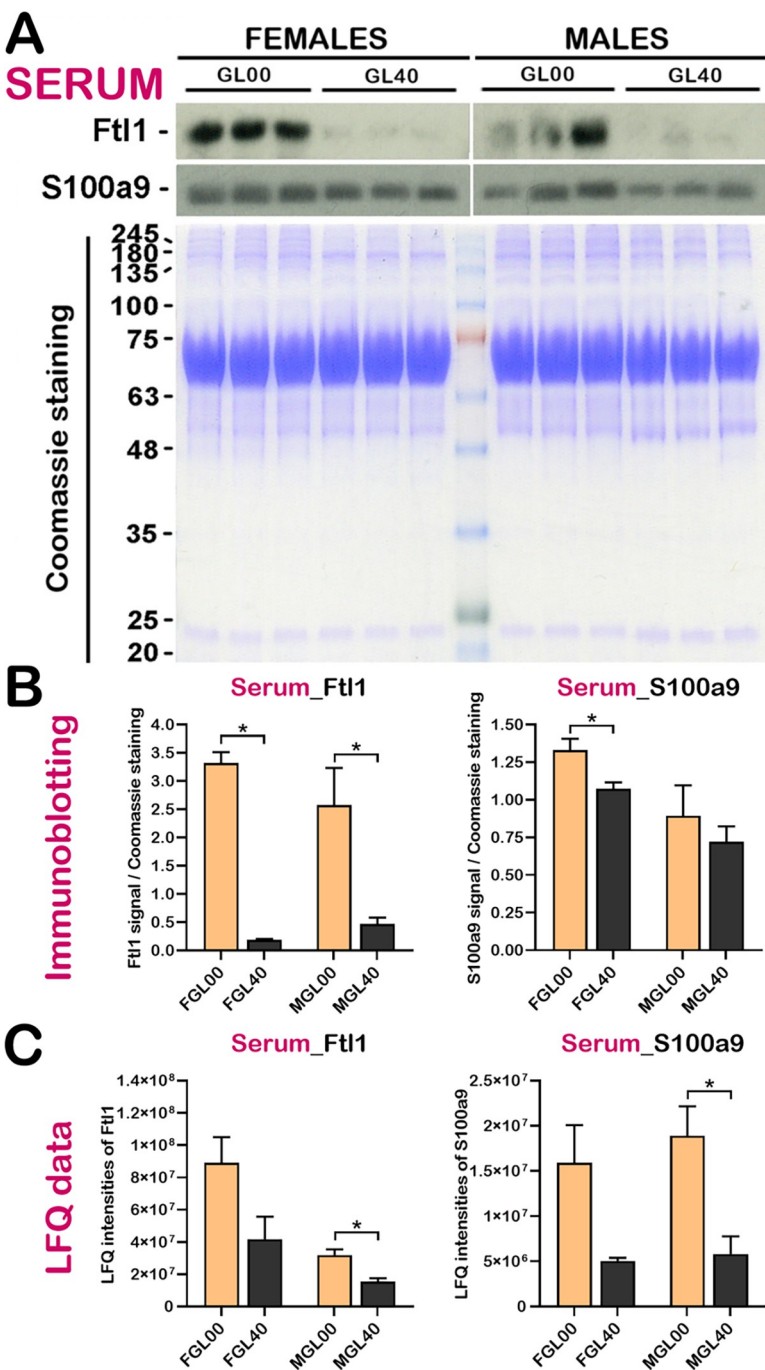

**Fig 12. Examination of Ftl1 and S100a9 levels in the serum of *Gulo⁻/⁻* mice.** (A) Western blot analysis of Ftl1 and S100a9 proteins in serum samples for vitamin C deficient mice (GL00) and 0.4% vitamin C treated *Gulo⁻/⁻* mice since weaning (GL40). Coomassie staining was used as in-gel loading control. (B) Signal quantification of Ftl1 and S100a9 over Coomassie staining signal from the immunoblot analysis in A (mean ± SEM). (C) Histograms present the LFQ data (mean ± SEM) for Ftl1 and S100a9 proteins in serum samples (N = 3 females and 3 males for each cohort; * *p*-value < 0.05; Welch's *t*-test).

The most important observation of the present study is that the proteome response profile to decreasing serum vitamin C levels was not only sex-dependent but also tissue specific. This was evident by the number and different proteins that significantly correlated positively or

inversely with serum vitamin C levels in the four organs under study and the specific biological processes that were modulated by vitamin C in each organ in either *Gulo*<sup>-/-</sup> females or males. More proteins significantly correlated with serum vitamin C concentrations in the heart, liver, and spleen of females compared to males. In contrast, more proteins correlated with vitamin C in the brain of males compared to female brain (Fig 5).

GO analysis of brain proteins correlating with serum vitamin C showed a significant positive correlation between proteins involved in lipid and glycolytic processes and serum vitamin C levels in the males. There was also a significant negative correlation between proteins of the mitochondrial complex I of the electron transport chain and vitamin C levels in males. These results suggest that the production of energy (as ATP) was altered upon vitamin C deficiency in the brain of *Gulo*<sup>-/-</sup> males. Such processes were not identified as significantly affected by vitamin C after Bonferroni adjustments in the brain of females. Moreover, even though all these proteins were also identified in several other organs in our mass spectrometry analyses, the glycolytic process and the proteins of the mitochondrial complex I were not correlating significantly with serum vitamin C levels in these other organs. Interestingly, a previous proteomic study on the brain of gonadectomized mice treated with sex hormone therapy revealed that electron transport chain-associated pathways were regulated by sex-hormone interactions [29]. Furthermore, other studies revealed higher basal ascorbate levels in the brain of male rats and that the loss of the female sex hormones increased significantly ascorbate concentrations in female brain tissues to levels comparable to the males [reviewed in [30]]. Additional work postulated that the enhanced ascorbate level in different parts of the male brain was to compensate for an augmented oxidative stress in males compared to females [30]. Noteworthy, the distribution and levels of vitamin C in the region of the forebrain, the hippocampus, and the cerebellum was found to differ in various pre-clinical animal models between sex groups [30, 31]. A limitation of our study is that mass spectrometry analyses were performed on the whole neocortex as we did not separate different brain regions for the analysis. Nevertheless, a vitamin C deficiency in the male brain has a greater impact on the abundance of proteins involved in bioenergetic processes than in females as observed with our *Gulo*<sup>-/-</sup> mouse cohort. Of relevance, previous comparative analyses on mitochondria from female and male brain tissues indicated differential mitochondria functions between sexes not only in female rodent models but also in premenopausal women [30]. The mitochondria in the brain of male generated more reactive oxygen species compared to the mitochondria in the brain tissues of females thus requiring more vitamin C for neuroprotective purposes in males [30]. Thus, the results obtained on the brain tissues of *Gulo*<sup>-/-</sup> mice provide pertinent and complementary information to further our understanding of the biological processes altered upon hypovitaminosis C (a vitamin C deficiency without a scorbutic phenotype) in females and males.

Few biological processes were affected in the cardiac tissue of vitamin C deficient *Gulo*<sup>-/-</sup> mice. Proteins involved in actin cytoskeleton organization (and thus sarcomere organization) were inversely correlated with serum vitamin C levels in the heart of females. In the first study of *Gulo*<sup>-/-</sup> mice in 2000, Maeda and colleagues reported that vitamin C deficiency had a deleterious effect on the architectural integrity of the large vessels in *Gulo*<sup>-/-</sup> mice [16]. Although we did not perform proteomic analysis on the aorta, the mass spectrometry data suggest that cardiomyocytes, which are the prevalent cell type in the heart tissue, would also exhibit some cellular architecture disorganization. However, actin cytoskeleton organization was not a biological process that significantly correlated with serum vitamin C in the heart of males. This sexual difference in the cardiac proteome is unclear at the present moment. Nevertheless, several studies have indicated the importance of vitamin C in the differentiation and maturation of cardiomyocytes including a better sarcomeric organization in these cells at least *in vitro* [32–34]. Thorough electron microscopic examination of cardiac tissue in both vitamin C

deficient *Gulo*-/- females and males will be required to determine the integrity of the cytoskeletal and sarcomere organization in cardiomyocytes of these mice.

GO analyses on the identified proteins correlating with serum vitamin C levels in the liver showed more biological processes modulated by vitamin C in females than in males. Interestingly, the hepatic tissue of *Gulo*-/- females exhibited a positive correlation between serum vitamin C and various proteins mainly involved in lipogenesis. In contrast, the liver of *Gulo*-/- males showed an inverse correlation between serum vitamin C and several proteins associated with lipid or fatty acid catabolism. Thus, fatty acid metabolic process (GO:0006631) has been significantly enriched in both females and males in the liver but showed different correlation with the vitamin C between the two sexes. Detailed examination of the proteins associated with this biological process revealed that the list of proteins in females (16 total) and the one from males (12 total) are different. This divergence probably explains the observed opposite correlation for fatty acid metabolic process in females and in males. Nevertheless, the overall outcomes of the regulation of these various proteins related to fatty acid metabolism tend toward a potential decreased accumulation of lipids under a vitamin C deficient condition in both females and males. Accordingly, a recent proteomic study indicated that sex hormone signaling plays a key role in lipid metabolism pathways and induces sexually dimorphic changes in the mouse liver proteome with a similar phenotypic outcome [35].

Vitamin C depletion has been associated with changes in lipid metabolism in other animal models deficient for this vitamin. SMP30 knockout mice that cannot synthesize vitamin C on their own like *Gulo*-/- mice have been shown to exhibit an impairment of *de novo* lipogenesis [36]. Additionally, the repercussion of altered lipid or fatty acid metabolism upon vitamin C depletion can be seen in experimental pre-clinical models of vitamin C deficiency under high fat diet. SMP30 knockout mice exhibit abnormal accumulation of hepatic cholesterol [36]. Guinea pigs, which are also considered *Gulo* null animals, treated with different high fat and low vitamin C diets revealed that a poor vitamin C status delays the reversion of a fatty liver condition toward a healthier hepatic phenotype [37]. Although there are contradictory epidemiological results regarding the improvement of liver functions by a vitamin C supplementation in subjects with metabolic dysfunction associated fatty liver disease (MAFLD), nonalcoholic fatty liver disease (NAFLD), and nonalcoholic steatohepatitis (NASH) [38], hypovitaminosis C has been associated with these chronic liver diseases [39–41].

In addition to altered lipid and fatty acid metabolism, various proteins of the acute phase response were inversely correlated with serum vitamin C and many subunits of the mitochondrial complex III of the electron transport chains were correlating positively with serum vitamin C in both *Gulo*-/- males and females. Of relevance, recent studies indicated that dysfunctional mitochondria participate in the aggravation of NAFLD [42]. Furthermore, it has been reported that the progression of NAFLD to non-alcoholic steatohepatitis is accompanied by a decrease mitochondrial respiratory chain including a 70% decrease of complex III activity in human patients [43]. Importantly, many proteins of the biological processes that were altered in the liver of vitamin C deficient *Gulo*-/- mice were also detected by mass spectrometry in the brain, the heart, and the spleen. However, such processes were not identified as significantly correlating with serum vitamin C levels in these other organs. We infer that vitamin C modulated the proteins of these biological processes in a tissue specific manner. It is noteworthy to mention that studies on the half-life of mitochondrial proteins of the respiratory chain complex (including complex III) has been reported to be shorter in the liver (half-life in the order of a week) than in the brain or the heart tissues (more than a month) [44, 45]. A vitamin C deficiency in *Gulo*-/- mice may have increased the already rapid turnover rate of the mitochondrial complex III proteins in hepatic tissue compared to the other organs under study. Accordingly, a recent work has indicated that the lower abundance of several proteins of the

mitochondrial complex III in the liver of vitamin C deficient *Gulo*$^{-/-}$ mice was occurring at a post-transcriptional level [46]. Of note, the precise pathogenesis of NAFLD progression under hypovitaminosis C conditions in human subjects, including altered lipid metabolism, oxidative stress, and inflammatory response is still unclear. The present work provides potential clues on enzymes in the liver that are affected by a vitamin C deficiency and that are potentially involved in disease progression.

The spleen is the largest immune tissue in the body and plays a vital role in immune defense due to the presence of lymphocytes and macrophages. It is a principal source of immunocyto-kines that modulate immune system responses. The spleen is also the major site of erythropoi-esis to meet increased demand for red cell production. The intracellular vitamin C levels in lymphocytes, monocytes (precursors of macrophages), and neutrophils have been reported to be in the millimolar ranges (1.5 to 3.5 mM) and entirely depends on vitamin C availability in the serum (which is in the micromolar range) [47]. GO analyses on the identified proteins cor-relating with serum vitamin C levels in the spleen showed more biological processes modu-lated by vitamin C in females than in males. Sexual dimorphism in the immune response has been well demonstrated in humans. Women exhibit lower infection rates than men for a vari-ety of bacterial, viral, and parasitic pathogens [48]. Although not entirely understood, the female sex hormones estrogen and progesterone, as well as the male androgens, such as testos-terone, elicit direct effects on the function and inflammatory capacity of immune cells [48]. *Gulo*$^{-/-}$ females exhibited an inverse correlation between serum vitamin C levels and biological processes related to cytoplasmic translation, ribosomal biogenesis and assembly, ubiquitin, protein ligase activity, DNA replication and repair. Based on our proteomic data, it is unknown which cell types in the spleen are the most affected by the vitamin C deficiency in the females. However, previous studies have reported a compensatory splenic erythropoiesis in vitamin C deficient adult *Gulo*$^{-/-}$ mice [49]. Stem cells of the erythroid lineage were signifi-cantly increased in vitamin C depleted two months old *Gulo*$^{-/-}$ mice [50]. Nevertheless, vitamin C depleted *Gulo*$^{-/-}$ mice could not overcome hemolytic anemia [49]. The increase abundance of proteins involved in translation, ribosomal biogenesis, and DNA replication/repair maybe a response of such erythroid cells upon vitamin C withdrawal from the diet in the spleen of four months old *Gulo*$^{-/-}$ females. Flow cytometry analyses of splenocytes from *Gulo*$^{-/-}$ mice will be required to isolate specific cell types for appropriate experiments to complement our observations.

*Gulo*$^{-/-}$ females also exhibited a positive correlation between serum vitamin C levels and actin cytoskeleton organization, cell-matrix adhesion, and platelet aggregation. Previous histological analysis of *Gulo*$^{-/-}$ mice revealed a severe disruption of the architecture of the splenic tissue after a five-week vitamin C withdrawal, though the study did not mention whether there was a difference between males and females [17]. It is worth noting that the regulation of actin cytoskeleton, phagocytosis, and the innate immune response system are major pathways regulated by members of the ten eleven translocation (TET) dioxygenases [47]. TET enzymes are vitamin C dependent DNA demethylases that will affect gene expres-sion in various immune cells [47, 50]. We have no evidence that the abundance of TET enzymes varies in the spleen of *Gulo*$^{-/-}$ females, but a depletion of vitamin C will affect their enzymatic activities. Further experiments on isolated splenocytes are required to quantify the levels of DNA methylation of genes for which protein levels are altered in vitamin C deficient *Gulo*$^{-/-}$ females.

For *Gulo*$^{-/-}$ males, mice exhibited a positive correlation between serum vitamin C levels and apolipoproteins involved in various processes of cholesterol transport and metabolism (includ-ing high-density lipoprotein (HDL) particle remodeling and negative regulation of very-low-density lipoprotein (VLDL) particle remodeling) in the spleen samples. Although the

abundance of apolipoproteins Apoa2, Apoa4, and Apoc3 in the spleen of *Gulo*$^{-/-}$ females correlated positively with serum vitamin C levels, they were not identified as part of a gene ontology term significantly altered by serum vitamin C concentrations after Bonferroni adjustment. Nevertheless, these results are consistent with human studies indicating increased changes in specific apolipoprotein A levels in subjects supplemented with vitamin C [51, 52]. Of note, the lipoprotein and lipid content of HDL and VLDL as well as their oxidation status will determine the anti-inflammatory or atherogenic impact of these particles *in vivo* [51, 52]. Interestingly, although many proteins that were identified in the spleen of *Gulo*$^{-/-}$ females and males were also quantified in the other organs under study, the biological processes identified in the spleen were not significantly recognized in the other organs with the DAVID bioinformatics tool. Nonetheless, the serum proteomic data indicated that proteins associated with high-density lipoprotein particle remodelling were positively correlated with serum vitamin C levels in both *Gulo*$^{-/-}$ females and males. Finally, serum proteins involved in the negative regulation of peptidase activity and the acute-phase response were inversely correlated to serum vitamin C levels in both females and males.

Using a robust statistical filtering strategy, we found that different biological processes were affected by vitamin C deficiency in the various organs of both females and males. However, several proteins that were not significantly associated with specific gene ontology terms could nonetheless correlate with serum vitamin C levels in multiple tissues. We thus determined which specific genes exhibited similar protein level alterations in several organs of *Gulo*$^{-/-}$ mice treated with different levels of vitamin C in drinking water. Few proteins correlated significantly with serum vitamin C levels (with a Spearman *p*-value < 0.05 and a with at least a 1.5-fold difference in abundance between GL00 and GL40 groups) in at least three of the four organs under study. Thirteen and four proteins correlated negatively and positively with serum vitamin C, respectively, in *Gulo*$^{-/-}$ females. The C3 and Serpina3k proteins correlated negatively and positively with serum vitamin C, respectively, in *Gulo*$^{-/-}$ males. Interestingly, GO analysis indicated that several proteins inversely correlating with serum vitamin C levels in at least three organs of the females were involved in the innate immune response. A similar tendency (not significant in all tissues) for an inverse correlation for many of these proteins were also observed in the male organs under study. They include Lrg1, Rbm3, Fetub, Hpx, and Cfi. Although a vitamin C deficiency would have damaged various cell types in different ways in the organs under study, a common reaction in these organs involved an innate immune response. Several studies in humans and rodents have indicated that a vitamin C deficiency results in impaired immunity and low chronic inflammation that can damage tissue with time [53].

Spearman analyses also indicated a positive correlation between the abundance of the ferritin complex (Ftl1 and Fth1) and serum vitamin C levels in *Gulo*$^{-/-}$ mice. Ferritin is involved in intracellular sequestering of iron ion. Of relevance, vitamin C regulates iron metabolism by increasing ferritin synthesis, inhibiting lysosomal ferritin degradation, and decreasing cellular iron efflux [54]. Vitamin C is thus required for iron absorption and homeostasis. Noteworthy, the abundance of serum Ftl1 protein correlated negatively with serum vitamin C levels, while Ftl1 abundance correlated positively with serum vitamin C in the tissues of *Gulo*$^{-/-}$ mice. Interestingly, some studies have suggested that ferritin can function as an iron carrier in the serum to distribute ferrous iron to cells in tissues [55].

To conclude, a major finding of this work indicates that several biological processes affected by a vitamin C deficiency may not only be sex specific dependent but also tissue specific dependent, even though many proteins have been identified and quantified in more than one organ.

## Supporting information

**S1 Fig. Label-free quantification after normalization of the data enabling comparison of samples and experimental groups.** Box plot depicting the distribution of each individual sample after normalization of the LFQ intensities using MaxQuant (N = 18 females and N = 18 males) in each organ.
(TIF)

**S2 Fig. Examination of the variation among the different biological brain replicates within the same experimental groups using multi-scatter plots analyses.** Pearson correlation coefficients obtained for each two-by-two comparison are indicated in blue on each graph.
(TIF)

**S3 Fig. Examination of the variation among the different biological heart replicates within the same experimental groups using multi-scatter plots analyses.** Pearson correlation coefficients obtained for each two-by-two comparison are indicated in red on each graph.
(TIF)

**S4 Fig. Examination of the variation among the different biological liver replicates within the same experimental groups using multi-scatter plots analyses.** Pearson correlation coefficients obtained for each two-by-two comparison are indicated in green on each graph.
(TIF)

**S5 Fig. Examination of the variation among the different biological spleen replicates within the same experimental groups using multi-scatter plots analyses.** Pearson correlation coefficients obtained for each two-by-two comparison are indicated in orange on each graph.
(TIF)

**S6 Fig. Label-free quantification of Ftl1, S100a9, Uqcrc1, and Uqcrfs1 proteins in brain, heart, liver, and spleen samples.** The histograms present the LFQ data for Ftl1, S100a9, Uqcrc1, and Uqcrfs1 proteins in the four different tissue lysates (N = 3 females and 3 males for each cohort; $*$ $p$-value $< 0.05$; Welch's $t$-test).
(TIF)

**S1 Table. Vitamin C measurements in the serum of female and male mice under study.**
(XLSX)

**S2 Table. Label-free LC-MS/MS data obtained for female brain, heart, liver, and spleen samples.**
(XLSX)

**S3 Table. Label-free LC-MS/MS data obtained for male brain, heart, liver, and spleen samples.**
(XLSX)

**S4 Table. Lists of proteins common and specific to the four organs for females and males.**
(XLSX)

**S5 Table. Biological processes associated with the lists of proteins that are common or specific to the four organs of females (GO terms highlighted in yellow are common with males).**
(XLSX)

**S6 Table. Biological processes associated with the lists of proteins that are common or specific to the four organs of males (GO terms highlighted in yellow are common with females).**
(XLSX)

**S7 Table. The lists of proteins for which LFQ intensities correlated with serum vitamin C levels in the four different organs for the females.**
(XLSX)

**S8 Table. The lists of proteins for which LFQ intensities correlated with serum vitamin C levels in the four different organs of males.**
(XLSX)

**S9 Table. Biological processes associated with proteins that correlated with serum vitamin C levels in the brain of females and males.**
(XLSX)

**S10 Table. Lists of proteins for each section of the Venn diagrams shown in Fig 7A.**
(XLSX)

**S11 Table. The lists of serum proteins for which LFQ intensities correlated with serum vitamin C levels in females and males.**
(XLSX)

**S12 Table. Biological processes associated with the lists of serum proteins correlating with serum vitamin C in females and males.**
(XLSX)

**S1 Raw image. Original uncropped and unadjusted blot/gel images.**
(PDF)

**S1 Graphical abstract.**
(TIF)

## Acknowledgments

We are grateful to Sylvie Bourassa from the Proteomics Platform of the Centre de Recherche du CHU de Québec, (Québec City, Canada) for the label-free liquid chromatography-tandem mass spectrometry analysis of all the samples.

## Author Contributions

**Conceptualization:** Lucie Aumailley, Michel Lebel.

**Data curation:** Lucie Aumailley, Michel Lebel.

**Formal analysis:** Lucie Aumailley, Michel Lebel.

**Funding acquisition:** Michel Lebel.

**Investigation:** Lucie Aumailley, Michel Lebel.

**Methodology:** Lucie Aumailley, Michel Lebel.

**Project administration:** Michel Lebel.

**Resources:** Michel Lebel.

**Software:** Lucie Aumailley, Michel Lebel.

**Supervision:** Michel Lebel.

**Validation:** Lucie Aumailley.

**Visualization:** Lucie Aumailley, Michel Lebel.

**Writing – original draft:** Lucie Aumailley, Michel Lebel.

**Writing – review & editing:** Lucie Aumailley, Michel Lebel.

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
