## [Decision Letter · Decision Letter 0]

19 Sep 2024

PONE-D-24-25338Sex and organ specific proteomic responses to vitamin C deficiency in the brain, heart, liver, and spleen of Gulo-/- micePLOS ONE

Dear Dr. Lebel,

Thank you for submitting your manuscript to PLOS ONE. After careful consideration, we feel that it has merit but does not fully meet PLOS ONE’s publication criteria as it currently stands. Therefore, we invite you to submit a revised version of the manuscript that addresses the points raised during the review process.

Please submit your revised manuscript by Nov 03 2024 11:59PM. If you will need more time than this to complete your revisions, please reply to this message or contact the journal office at plosone@plos.org. Please include the following items when submitting your revised manuscript:A rebuttal letter that responds to each point raised by the academic editor and reviewer(s). You should upload this letter as a separate file labeled 'Response to Reviewers'.A marked-up copy of your manuscript that highlights changes made to the original version. You should upload this as a separate file labeled 'Revised Manuscript with Track Changes'.An unmarked version of your revised paper without tracked changes. You should upload this as a separate file labeled 'Manuscript'.If applicable, we recommend that you deposit your laboratory protocols in protocols.io to enhance the reproducibility of your results. Protocols.io assigns your protocol its own identifier (DOI) so that it can be cited independently in the future. For instructions see: https://journals.plos.org/plosone/s/submission-guidelines#loc-laboratory-protocols. Additionally, PLOS ONE offers an option for publishing peer-reviewed Lab Protocol articles, which describe protocols hosted on protocols.io. Read more information on sharing protocols at https://plos.org/protocols?utm_medium=editorial-email&utm_source=authorletters&utm_campaign=protocols.

We look forward to receiving your revised manuscript.

Kind regards,

Swaroop Kumar Pandey

Academic Editor

PLOS ONE

 1. When submitting your revision, we need you to address these additional requirements. Please ensure that your manuscript meets PLOS ONE's style requirements, including those for file naming. The PLOS ONE style templates can be found at  https://journals.plos.org/plosone/s/file?id=wjVg/PLOSOne_formatting_sample_main_body.pdf and  https://journals.plos.org/plosone/s/file?id=ba62/PLOSOne_formatting_sample_title_authors_affiliations.pdf. 

 2. PLOS ONE now requires that authors provide the original uncropped and unadjusted images underlying all blot or gel results reported in a submission’s figures or Supporting Information files. This policy and the journal’s other requirements for blot/gel reporting and figure preparation are described in detail at https://journals.plos.org/plosone/s/figures#loc-blot-and-gel-reporting-requirements and https://journals.plos.org/plosone/s/figures#loc-preparing-figures-from-image-files. When you submit your revised manuscript, please ensure that your figures adhere fully to these guidelines and provide the original underlying images for all blot or gel data reported in your submission. See the following link for instructions on providing the original image data: https://journals.plos.org/plosone/s/figures#loc-original-images-for-blots-and-gels.    In your cover letter, please note whether your blot/gel image data are in Supporting Information or posted at a public data repository, provide the repository URL if relevant, and provide specific details as to which raw blot/gel images, if any, are not available. Email us at plosone@plos.org if you have any questions.  

 3. Thank you for stating the following financial disclosure:   [This work was supported by the Canadian Institutes of Health Research, Canada (PJT-173398) to Michel Lebel.].   Please state what role the funders took in the study.  If the funders had no role, please state: ""The funders had no role in study design, data collection and analysis, decision to publish, or preparation of the manuscript.""  If this statement is not correct you must amend it as needed.  Please include this amended Role of Funder statement in your cover letter; we will change the online submission form on your behalf.   4. Thank you for stating the following in the Acknowledgments Section of your manuscript:  [We are grateful to Sylvie Bourassa from the Proteomics Platform of the Centre de Recherche du CHU de Québec, (Québec City, Canada) for the label-free liquid chromatography-tandem mass spectrometry analysis of all the samples. This work was supported by the Canadian Institutes of Health Research, Canada (PJT-173398) to M.L.] We note that you have provided funding information that is not currently declared in your Funding Statement. However, funding information should not appear in the Acknowledgments section or other areas of your manuscript. We will only publish funding information present in the Funding Statement section of the online submission form.  Please remove any funding-related text from the manuscript and let us know how you would like to update your Funding Statement. Currently, your Funding Statement reads as follows:   [This work was supported by the Canadian Institutes of Health Research, Canada (PJT-173398) to Michel Lebel.]   Please include your amended statements within your cover letter; we will change the online submission form on your behalf.

Reviewers' comments:

Reviewer's Responses to Questions

**Comments to the Author**

1. Is the manuscript technically sound, and do the data support the conclusions?

Reviewer #1: Yes

Reviewer #2: Yes

2. Has the statistical analysis been performed appropriately and rigorously? 

Reviewer #1: Yes

Reviewer #2: Yes

3. Have the authors made all data underlying the findings in their manuscript fully available?

Reviewer #1: Yes

Reviewer #2: Yes

4. Is the manuscript presented in an intelligible fashion and written in standard English?

Reviewer #1: Yes

Reviewer #2: Yes

5. Review Comments to the Author

Reviewer #1: The recent manuscript describes the differential protein expression in different tissues of male and female mice with the correlation of vitamin C concentration. The study conclude that the the Proteome profile to decreasing serum vitamin C level is not only sex-dependent but also tissue specific. The manuscript is well written, and results are clearly presented and also recommended for the publication.

Specific comment:

The biological process which are similar in male and female are also showing different correlation with the vitamin C, explain the probable reason behind the same.

Reviewer #2: Authors have presented a study where they have found out that Veitamin C concentrations in male and female mice differentially modulates the abundance of various proteins in hepatic tissues. To support their study, the authors have performed LC-MS/MS to identify and quantify proteins that correlate with serum vitamin C concentrations in thewhole brain, heart, liver, and spleen tissues in mice deficient for the enzyme LGulonolactone oxidase required for vitamin C synthesis in mammals. Their work shows that Vitamin C deficiency affect numerous biological processes that are tissue and sex specific dependent. Their study provides a preliminary comprehensive analysis of various proteins that significantly correlate with Vitamin C levels. This information can be useful for the Science community and can add to some therapeutic advancements in the longer run.

6. PLOS authors have the option to publish the peer review history of their article (what does this mean?). If published, this will include your full peer review and any attached files.

Reviewer #1: No

Reviewer #2: **Yes: **Dr. Aayushi Singh

---

## [Author Response · Author response to Decision Letter 0]

24 Sep 2024

Reviewer #1

Reviewer’s Comments

The recent manuscript describes the differential protein expression in different tissues of male and female mice with the correlation of vitamin C concentration. The study concludes that the Proteome profile to decreasing serum vitamin C level is not only sex-dependent but also tissue specific. The manuscript is well written, results are clearly presented, and also recommended for the publication.

Specific comment:

The biological process which are similar in male and female are also showing different correlation with the vitamin C, explain the probable reason behind the same. 

Response: As requested by reviewer #1, we explained in the discussion section the reason why the similar biological process correlated in opposite direction between females and males upon vitamin C deficiency. We added the following information in the text and introduced an additional reference. 

GO analyses on the identified proteins correlating with serum vitamin C levels in the liver showed more biological processes modulated by vitamin C in females than in males. Interestingly, the hepatic tissue of Gulo-/- females exhibited a positive correlation between serum vitamin C and various proteins mainly involved in lipogenesis. In contrast, the liver of Gulo-/- males showed an inverse correlation between serum vitamin C and several proteins associated with lipid or fatty acid catabolism. Thus, fatty acid metabolic process (GO:0006631) has been significantly enriched in both females and males in the liver but showed different correlation with the vitamin C between the two sexes. Detailed examination of the proteins associated with this biological process revealed that the list of proteins in females (16 total) and the one from males (12 total) are different. This divergence probably explains the observed opposite correlation for fatty acid metabolic process in females and in males. Nevertheless, the overall outcomes of the regulation of these various proteins related to fatty acid metabolism tend toward a potential decreased accumulation of lipids under a vitamin C deficient condition in both females and males. Accordingly, a recent proteomic study indicated that sex hormone signaling plays a key role in lipid metabolism pathways and induces sexually dimorphic changes in the mouse liver proteome with a similar phenotypic outcome [35].

The new reference 35 is as follows:

35. Harney DJ, Cielesh M, Roberts GE, Vila IK, Viengkhou B, Hofer MJ, et al. Dietary restriction induces a sexually dimorphic type I interferon response in mice with gene-environment interactions. Cell Rep. 2023;42(6):112559. Epub 2023/05/27. doi: 10.1016/j.celrep.2023.112559. PubMed PMID: 37243595.

Reviewer #2

Authors have presented a study where they have found out that Veitamin C concentrations in male and female mice differentially modulates the abundance of various proteins in hepatic tissues. To support their study, the authors have performed LC-MS/MS to identify and quantify proteins that correlate with serum vitamin C concentrations in thewhole brain, heart, liver, and spleen tissues in mice deficient for the enzyme LGulonolactone oxidase required for vitamin C synthesis in mammals. Their work shows that Vitamin C deficiency affect numerous biological processes that are tissue and sex specific dependent. Their study provides a preliminary comprehensive analysis of various proteins that significantly correlate with Vitamin C levels. This information can be useful for the Science community and can add to some therapeutic advancements in the longer run.

Response: Thank you.

---

## [Editor Report · Decision Letter 1]

26 Sep 2024

Sex and organ specific proteomic responses to vitamin C deficiency in the brain, heart, liver, and spleen of Gulo-/- mice

PONE-D-24-25338R1

Dear Dr. Michel Lebel,

We’re pleased to inform you that your manuscript has been judged scientifically suitable for publication and will be formally accepted for publication once it meets all outstanding technical requirements.

Kind regards,

Swaroop Kumar Pandey

Academic Editor

PLOS ONE

---

## [Editor Report · Acceptance letter]

1 Oct 2024

PONE-D-24-25338R1 

PLOS ONE

Dear Dr. Lebel, 

I'm pleased to inform you that your manuscript has been deemed suitable for publication in PLOS ONE. Congratulations! Your manuscript is now being handed over to our production team.

Kind regards, 

on behalf of

Dr. Swaroop Kumar Pandey 

Academic Editor

PLOS ONE